# The critical role of uncertainty in projections of hydrological extremes

Hadush K. Meresa and Renata J. Romanowicz

Department of Hydrology and Hydrodynamics, Institute of Geophysics, Polish Academy of Sciences, Warsaw, Poland

*Correspondence to*: Renata J. Romanowicz (romanowicz@igf.edu.pl)

**Abstract.** This paper aims to quantify the uncertainty in projections of future hydrological extremes in the Biala Tarnowska River at Koszyce gauging station, south Poland. The approach followed is based on several climate projections obtained from the EURO-CORDEX initiative, raw and bias-corrected realizations of catchment precipitation, and flow simulations derived using multiple hydrological model parameter sets. The projections cover the 21$^{st}$ century. Three sources of
uncertainty are considered: one related to climate projection ensemble spread, the second related to the uncertainty in hydrological model parameters and the third related to the error in fitting theoretical distribution models to annual extreme flow series. The uncertainty of projected extreme indices related to hydrological model parameters was conditioned on flow observations from the reference period using the Generalised Likelihood Uncertainty Estimation (GLUE) approach, with separate criteria for high and low flow extremes. Extreme (low and high) flow quantiles were estimated using the
Generalized Extreme Value (GEV) distribution at different return periods and were based on two different lengths of the flow time series. A sensitivity analysis based on the Analysis of Variance (ANOVA) shows that the uncertainty introduced by the hydrological model parameters can be larger than the climate model variability and the distribution fit uncertainty for the low-flow extremes whilst for the high-flow extremes higher uncertainty is observed from climate models than from hydrological parameter and distribution fit uncertainties. This implies that ignoring one of the three uncertainty sources
may cause great risk to future hydrological extreme adaptations and water resource planning and management.

## 1 Introduction

Hydrological models are useful in water resources planning and management, flood and drought prediction, assessments of catchment-scale impacts of climate change, and the understanding of system dynamics. In particular, coupling of hydrological models and climate models is important in understanding the influence of climate changes on low and high
flows (Lawrence and Hisdal, 2011; Meresa et al., 2016). Research on the impact of climate changes on future hydrological extremes is usually performed by an application of hydrological models to the projected meteorological inputs under assumed future climate scenarios (Wilby and Harris, 2006; Honti et al., 2014). The standard procedure consists of a chain of consecutive actions, starting from the choice of a general circulation model (GCM) driven by an assumed greenhouse

gas emission scenario, through downscaling of climatic forcing to a catchment scale, e.g., using the regional climate model (RCM), hydrological modelling and estimation of hydrological extreme indices using statistical tools. Each of the serially linked processes involves uncertainties that propagate through the computational pathway. Among many possible sources of uncertainty, the main sources are the uncertainties related to future climate scenarios, climate models, downscaling techniques and hydrological modelling. We cannot directly assess the impact of the first three sources of uncertainties on predictions of hydrological extremes in the future due to a lack of observations of future climate realizations. This is one of the reasons why the term projections is used instead of predictions (Honti et al., 2014). Therefore these first three sources of uncertainty have an "epistemic" nature and cannot be decreased (Beven, 2016). On the other hand, the introduction of conditioning based on available past observations of climatic and hydrological variables allows a decrease of the "predictive" hydrological model uncertainty to be achieved. The calibrated hydrological models are forced with climate projections derived from climate models. However, hydrological models that produce acceptable results for an observed baseline period may respond differently when forced with the climate change scenario (Gosling and Arnell, 2011; Thompson et al., 2013; Lespinas et al., 2014). For several decades interest in hydrological structural and parameter uncertainty has been increasing and has become an important part of modelling (Ouyang et al., 2014; Sellami et al., 2014; Osuch et al., 2015). It has been widely verified and acknowledged that different model structures and parameterizations can lead to similar responses and, thus, there are no unique structure and parameter sets for acceptable or behavioural hydrological model responses for reproducing the observation data (Beven, 2006; Poulin et al., 2011). In addition to the parameter and structural uncertainty (Poulin et al., 2011), the hydrological model is also exposed to uncertainty which arises from various sources not directly mentioned above, including interdependency among the climate models (Wilby and Harris, 2006; Ghosh and Katkar, 2012; Tian et al., 2016) or downscaling of GCM projections (Sunyer et al., 2015; Vormoor et al., 2015).

The issues of uncertainty in hydrological modelling and hydrological projections due to climate change are not new; there is much research published on this subject in global and regional studies (Todd et al., 2011; Addor et al., 2014, Abbaspour et al., 2015). However, few of the case studies at a catchment level were trying to assess the influence of uncertain future and hydrological parameter uncertainty (Poulin et al., 2011; Bennett et al., 2012; Stenschneider et al., 2012, 2015; Vormoor et al., 2015). In a number of studies the hydrological model structural and parametric errors are dealt with using a multi-model approach and introducing weights for hydrological model parameter sets following assumed goodness of fit criteria, e.g. in the form of a likelihood function (Wilby and Harris, 2006, Steinschneider et al., 2012, Addor et al., 2014). Addor et al. (2014) concentrated on the influence of different hydrological model structure, involving three hydrological models, emission scenarios, climate models, post-processing and catchments. Their results indicate that influence of model structure varies with the catchment. However the authors did not take into account hydrological model parameter uncertainty, which is included in the present paper. Following the results presented by Demirel et al. (2013a) the choice of the GCM/RCM has larger influence than the choice of the emission scenario on the projections of low flow indices. Similar

findings for the high-flow indices were presented by Osuch et al. (2016). There is general agreement that we cannot avoid uncertainty in climate models (Knutti and Sedlacek, 2012). The question arises as to how large the uncertainty is and if it is acceptable to the end-user in adaptations to climate change and flood and drought risk assessments.

In this study, we assess the critical role of uncertainty in the projection of future hydrological extremes in the Biala Tarnowska mountainous catchment in Poland in the 21$^{st}$ century. We consider three sources of uncertainty. These are epistemic climate projection uncertainty, hydrological model parameter uncertainty and uncertainty of extreme index estimates (the error in fitting theoretical distribution models to annual extreme flow series). We restricted the sources of epistemic uncertainty to that which have the largest impact – i.e. climate model spread, omitting the uncertainty related to bias correction and emission scenario. The error related to the distribution fit was included as an essential part of the extreme index evaluation, which requires extrapolation of annual maximum or minimum flow distributions to higher order quantiles (e.g. 1-in-100 year, or 1-in-200 year). Osuch et al. (2016) presented the influence of emission scenario, climate model, bias correction method and catchment on flow indices in a case study that included the same catchment, Biala Tarnowska. In this respect, our paper is an extension of that paper, focusing on the influence of hydrological model parameter uncertainty on annual maximum and minimum flow projections.

We apply a non-formal approach to estimate the uncertainty related to hydrological model parameters, namely, the Generalized Likelihood Uncertainty Estimation (GLUE) method of Beven and Binley (1992, 2016). The other sources of uncertainty are dealt with by means of a direct assessment of variability in extreme index estimates. Seven climate projections applied are derived from the high-resolution regional climate change ensemble within the World Climate Research Program Coordinated Regional Downscaling Experiment (EURO-CORDEX) initiative (Jacob et al., 2014). Two separate goodness-of-fit criteria are chosen to constrain the hydrological parameter uncertainty of high and low flow estimates. In this way, different parameter sets are chosen for the description of high and low flow catchment regimes. This approach does not eliminate the problem of parameter non-stationarity but helps to choose the model behaviour adequate to the flow regime. The uncertainty related to the distribution fit is analysed in two stages, using, separately, the 30-year long and 130-year long time series of future flow projections to derive the quantiles of maximum and minimum annual flows. The popular method of a comparison of changes in flow quantiles between the reference period and future periods is based on relatively short (e.g. 30-year) periods. It is well known that an extrapolation of a distribution function based on a 30-year long time series towards 1-in-100 year quantiles involves very large errors (Strupczewski et al., 2011). Even the estimates of 1-in-30 year quantiles based on the 30-year long data are biased with large errors. We compare these errors with those involved on 1-in-30 year quantile estimates obtained using the 130 year long time series. The question we pose is whether the estimates of future trends of extreme indices and their relative changes can be useful at all in view of the uncertainties involved.

The paper is organized into five sections. The second and third sections describe, respectively, the case study and the methodology applied. The fourth section presents the results and discussions of the uncertainty analysis and derived changes in future low and high flow extremes; the fifth section presents the conclusions.

## 2 Study area and hydro-climatic data

### 2.1 Study area and observed data characteristics

The Biala Tarnowska catchment, located in the mountainous part of Poland, was chosen as a case study. This catchment is one of the representative Polish catchments chosen following an extensive analysis of available hydro-meteorological and geomorphological data (Romanowicz et al., 2016a). The catchment area is about 967 km$^2$, with forests covering much of the upper elevations and the river is characterized by nearly-natural conditions. The location of the catchment is given in Fig. 1. Precipitation varies in intensity and duration over the catchment area. Observations from five gauging stations were used to derive areal precipitation in the catchment by means of Thiessen polygons. No elevation correction was applied in this study. However it was applied to the same catchment by Benninga et al (2016) and showed that the increase in precipitation due to the elevation is about 3%. The annual maximum precipitation, annual minimum streamflow and annual mean streamflow of the catchment were 68.3 mm, 0.4 m$^3$s$^{-1}$ and 5.43 m$^3$s$^{-1}$ respectively over the observation period 1971-2000.

Biala Tarnowska has a mixed (rainfall and snow-melt originated) flood regime. In this study, daily hydro-meteorological observations and estimated potential evapotranspiration were used as an input to the hydrological model HBV (Bergström, 1995). Observed hydrological and climate daily time series of precipitation, temperature and flow for 39 years from November 1970 to October 2010 were obtained from the National Water Resource and Meteorology Office (IMGW) in Poland. Daily potential evapotranspiration was calculated using the temperature based Hamon approach (Hamon, 1961). The daily flow data from the Koszyce Wielkie hydrological station for a period of 39 years (1971-2010) were used in the calibration (1971-2000) and validation (2001-2010) stages.

### 2.2 Future climate data

Daily temperature and precipitation projections were obtained from the EURO-CORDEX initiative project (http://www.eurocordex.net/) which provides regional climate projections at a spatial resolution of 12.5 km (EUR-11) for median (RCP45) emission scenario and covering the time period 1971-2100 (Kotlarski et al., 2014). This ensemble contains four different RCMs driven by three different GCMs. The names and model affiliations are given in Table 1. The RCP 4.5 was applied because it is a stabilization scenario and thus assumes the imposition of emissions mitigation policies. The RCP 4.5 is derived from its own "reference", or "no-climate-policy", scenario. This reference scenario is unique to RCP 4.5 and differs from RCP 8.5, RCP 6.0 and RCP 2.6 (Jacob et al., 2014). The influence of the emission scenario on

flood indices was studied by Osuch et al. (2016) whilst the low flows were analysed by Demirel et al. (2013a) and Osuch et al. (2017). Those studies indicated that the choice of emission scenario has a relatively small influence on the results as compared to climate model spread.

## 3 Methodology

### 3.1 Research approach

The approach applied here to the derivation of future projections of flow extremes follows the forward modelling chain (Wilby and Harris, 2006) and consists of the following steps: (i) choice of climate projections simulated using the ensemble of GCM/RCMs under the assumed carbon emission scenario (here RCP4.5) and dynamically downscaled to the catchment scale; (ii) bias correction of projected meteorological time series of temperature and precipitation; (iii) hydrological simulations of flow using raw and bias-corrected meteorological projections for a set of hydrological model parameters; (iv) derivation of extreme flow indices using empirical and distribution-based frequency analysis tools and two different lengths of time series (30 and 130 years) of the analysed flow extremes. The assessment of projection uncertainty is performed by running multiple simulations and evaluating the impact of each of the chain modules on the total uncertainty of the results (Wilby and Harris, 2006; Steinschneider et al., 2012).

### 3.2 Climate projections: bias correction

The downscaling of the GCM output using either statistical or dynamic (RCM) approaches does not take into account any feedback mechanisms existing within land-surface processes and therefore the meteorological projections can be biased (Falloon et al., 2014). Several studies have identified the need to check and correct bias, in the GCM/RCMs output, before its use in impact studies (Gudmundsson et al., 2012; Gutjahr and Heinemann, 2013; Teutschbein and Seibert, 2013; Teng et al., 2015). Most of those studies were focused on mean output values. Osuch et al. (2016) compared five different distribution-based Quantile Mapping (QM) techniques applied in the derivation of extreme flow indices (flow quantiles and mean annual maximum flow). Their results showed the single gamma distribution mapping to be the one which produced the observed characteristics most accurately of all the techniques and regions studied. However, the QM technique applied to simulated precipitation series in the reference period (1971-2000) may result in an alteration of the modelled maximum runoff (Ehret et al., 2012; Teng et al., 2015). On the other hand, the low extreme values require bias-corrected precipitation input due to the persistent and unrealistic drizzle present in raw precipitation data. The drizzle effect (i.e., too many days with very low precipitation intensity and too few dry days) is related to the performance of climate models. It presents itself in the form of frequent rainfall of a very small intensity. The physics behind precipitation generation is very complex and involves processes operating on a wide range of scales. The frequent 'drizzle' is produced mainly by convective parameterization. It appears in many climate models and invokes errors in the intensity and frequency

of precipitation (Maraun, 2013). A correction can be performed using the number of wet days in a month (Osuch et al., 2016). Because of this bias in precipitation, using direct climate model outputs as inputs to hydrological modelling for low flow analysis often leads to unrealistic results and therefore bias correction is required in the case of low flow projections. Temperature projections are corrected using the empirical QM approach.

## 3.3 Hydrological modelling

The HBV hydrological model version applied is based on Lindstrom et al. (1997) and is written in Matlab®. It is a lumped conceptual multi-reservoir-type model for daily runoff simulation from daily inputs (Lindstrom et al., 1997). The original MATLAB code from the Twente University, NL, was further developed and adjusted for the purpose of climate impact studies in the Institute of Geophysics PAS. The model uses precipitation, air temperature and potential evaporation data as

inputs. The HBV model has four main routines: (i) snow; (ii) soil moisture; (iii) fast response; and (iv) slow response routing. These routines are governed mainly by fourteen HBV parameters, of which, six (*TT, TTI, CFMAX, DTTM, CFR, WHC*), three (*FC, LP, BETA*), three (*KF, ALPHA, CFLUX*) and two (*KS, PERC*) parameters are representing each routine respectively. Not all HBV model parameters have significant impact on the simulated flows. The HBV model was applied in different hydro-climatic conditions by many researchers (e.g., Demirel et al., 2013b; Seibert and Vis, 2016,).

Romanowicz et al. (2013) discussed the most sensitive parameters of the HBV model for both high flow and low flow characteristics. Other studies of the HBV model parameter sensitivity were presented by Osuch (2015) and Osuch et al. (2015). The set of six most sensitive parameters for the extreme high and low flow conditions was chosen following those studies. These are *FC*, *BETA*, *LP*, *KS*, *CFMAX* and *PERC*. A full description of the HBV hydrological model which we used can be found in Osuch et al. (2015). Osuch et al. (2015) also compared three sensitivity analysis techniques to describe

the HBV model parameter interactions. The studies mentioned show that the sensitivity of HBV model parameters vary depending on the catchment characteristics and time period of model evaluation. The parameters chosen were the most sensitive in the wide range of input variability. The other HBV model parameters show small influence on model output independent of the event type. The application of different evaluation criteria for low and high flow conditioning allows for treating the parameters according to their influence on the flows, i.e. it takes into account model output sensitivity

(Saltelli et al. 2006). We used the output of Osuch et al. (2015) to eliminate less sensitive HBV model parameters in order to minimize computational cost.

Hydrological models are usually calibrated using the available observations under the assumption of stationarity of their parameters. Depending on the purpose of the modelling, different criteria may be used (Romanowicz et al., 2013). Usually, the research is aimed at finding the compromise of a model performance between high and low flow simulations.

The Nash-Sutcliffe criterion (*NSE*) (Nash and Sutcliffe, 1970) belongs to those most widely used. When based on the whole calibration observation series, it provides parameter sets that favour medium-to-high flows (Gupta et al., 2009). Deckers et al. (2010) applied different time periods of observations related to high and low flows. The authors used multi-

objective criteria that combined different aspects of model performance. However, we do not always need to look for a compromise in model performance when choosing the parameter sets of a model. Where hydrological extremes are concerned, the average model performance is not of interest. Rather, we want to obtain robust model performance for very low or very high flow values. Therefore, in this study we use in parallel two objective functions to encapsulate the high and low flow characteristics. The *NSE* criterion ($J_{NSE}$) is used here to calibrate the high-flow-oriented HBV model. The low-flow HBV model is calibrated using the *NSE* for the logarithm of flow $J_{logNSE}$. The criteria are defined as follows:

$$J_{NSE} = 1 - \frac{\sum_{t=1}^{T}(Q_{t,sim} - Q_{t,obs})^2}{\sum_{t=1}^{T}(Q_{t,obs} - \overline{Q_{obs}})^2}, \tag{1}$$

$$J_{logNSE} = 1 - \frac{\sum_{t=1}^{T}(\log(Q_{t,sim}) - \log(Q_{t,obs}))^2}{\sum_{t=1}^{T}(\log(Q_{t,obs}) - \overline{\log(Q_{obs})})^2}, \tag{2}$$

Where $Q_{t,sim}$ denotes simulated flow in time *t* (here days), *t=1,...,T*; $Q_{t,obs}$ denotes observed flow in time *t*; $\overline{Q_{t,obs}}$ denotes mean observed flow and $\overline{\log(Q_{t,obs})}$ denotes mean of logarithm of flows.

Depending on the formulation of the problem, either deterministic or stochastic methods can be used to derive a set of the best model parameters (Romanowicz and Macdonald, 2005). In this study we use a stochastic formulation and we apply the Generalized Likelihood Uncertainty Estimation GLUE approach of Beven and Binley (1992) to calibrate the HBV model and provide an estimation of the model parameter uncertainty.

## 3.4 Hydrological model parameter uncertainty

The GLUE approach is one of the non-formal statistical methods that involve direct Monte Carlo MC simulations. The entire parameter space is explored by running the model simulations for a large number of parameter combinations and evaluating the model response using some chosen goodness of fit criterion (Beven, 2007). The idea of an optimal system representation is rejected and the equifinality concept is accepted for the behavioural parameter sets.

Following that approach, the parameter space is sampled over the whole feasible range and the errors between simulated model results and observations are used to derive the parameter set weighting. The number of samples depends on the number of model parameters but also on the model computing times and it may vary from hundreds to hundreds of thousands (Beven and Binley, 2014). Many research papers recommend over 10000 MC simulations (Jin et al., 2010, Romanowicz et al., 2013, Houska et al., 2014). In this study we apply the version of GLUE that uses the behavioural parameter sets, defined by a threshold value of the selected criterion (Beven, 2006). The behavioural thresholds for both criteria are selected following the model performance in the calibration period. The choice of high threshold values results in narrow confidence limits of the predictions and (usually) a small behavioural parameter set. However, when the chosen threshold is too high, the 0.95 confidence limits do not include 95% of the observations. On the other hand, too low a threshold value will result in too wide confidence limits. Therefore it is important to choose the right threshold value. In

this work the threshold values are chosen by the "trial and error approach". The choice of two different criteria, one for high and one for low flow extremes, yields two different behavioural parameter sets describing model performance in two different (low- and high- flow) hydro-meteorological conditions.

## 3.5 Uncertainty related to fitting the Generalized Extreme Value Distribution GEV to extreme flow projections

The choice of the Generalized Extreme Value Distribution GEV (Coles, 2001) followed the validation of suitability of this distribution to describe the projected annual maximum and minimum flows using probability plots for the Biala Tarnowska. The GEV distribution model was applied to all the climate models and the a posteriori hydrological model parameter sets. The MATLAB-based GEV-fitting algorithm provides estimates of the median and the 0.95 confidence bands for the parameters of GEV distribution. These parameters were subsequently used to obtain lower and upper confidence bands of quantiles of extreme index distribution through the inverse GEV model (Coles, 2001, eq. 3.4). In order to simplify the procedure, instead of sampling from the GEV parameters within the parameter space common to all hydrologic and climate model simulations, we sampled from each set of parameters assuming a normal distribution with the variance specified by the GEV parameter lower and upper 0.95 confidence values, and in addition, assuming the independence of the GEV model parameters. The obtained 0.95 GEV distribution confidence values were used to estimate the spread of results related to the distribution fit. Bearing in mind that the aim of this study was to assess the ranges of uncertainty of extreme indices rather than their exact values, and the large number of simulations, it was not possible to choose among different distribution functions the best distribution for each projected time series.

## 3.6 Sensitivity analysis using ANOVA: variance decomposition

Many global sensitivity methods have been proposed and used, such as Fourier Amplitude Sensitivity Test (FAST), Regional Sensitivity Analysis (RSA), Analysis of Variance (ANOVA), Parameter Estimation Software (PEST), Morris, and Sobol method (Saltelli et al., 2006). Among these methods, ANOVA has proved to be one of the most robust and effective tools to analyze both continuous and discrete factors (Montgomery, 1997), and is widely applied in hydrology (Bosshard et al., 2013; Zhan et al., 2013; Lagerwalla et al., 2014; Addor et al., 2014; Giuntoli et al., 2015; Osuch, 2015). We used the ANOVA approach due to its numerical facility (MATLAB) and ability to evaluate the main and interactive effects between the factors considered. To identify the relative contribution of each source of uncertainty, corresponding to the parameter sets (*P*), climate models (*C*) and parameter distribution sets (*D*), from the spread of flow quantile change in the near and far future, we use the following ANOVA model:

$$T_{ijk} = \mu + P_i + C_j + D_k + (P + C)_{ij} + (P + D)_{ik} + (C + D)_{jk} + \varepsilon_{ijk}$$

(3)

Where: $T_{ijk}$ is a total sum squared error for the specific hydrological extreme indicator (e.g. relative change in the empirical high-flow quantile at 30-year return period $Q_{T30}$) for the $i^{th}$ parameter sets range, $j^{th}$ climate model and $k^{th}$ distribution parameter range and $\mu$ is the overall mean and $\varepsilon_{ijk}$ denotes the white Gaussian error.

### 3.7 Design of numerical experiments

We present here an assessment of the uncertainty in projected hydrological extremes for two different lengths of data periods. Firstly, the annual maximum and minimum (seven-day average) flow quantiles are derived for 30-year periods, the so-called near future (2021-2050), and far-future (2071-2100) and are compared with the reference period (1971-2000). Secondly, a frequency analysis of annual maximum and minimum flows is performed based on the whole 130 years of seven GCM/RCM projections for the period 1971-2100. Since the Biala Tarnowska flow projections do not show any non-
stationarity in extreme flow events (Meresa et al., 2017), it is possible to compare the uncertainty of estimates of extreme indices obtained from the 30-year long and 130-year long time series. It can be expected that the uncertainty of extreme flow quantiles will be larger for short time series, but we do not know how much larger it can be and therefore that comparison is not obvious. The comparison can help in answering our research question on how reliable is the approach commonly used in climate impact studies consisting of a comparison of 30-year based estimates of extreme flow indices
between reference and future periods.

As explained in the section 3.3, a stochastic formulation is applied to the estimation of the HBV model parameters. That means, 20000 simulations of the HBV model were run for the 30-year long calibration period (1971-2000) with parameters sampled randomly within the assumed parameter ranges (Table 2). That number was dictated by the practical requirement of dealing with not too large data files. The parameter ranges were chosen following the results of deterministic
optimisation performed earlier and reported by Romanowicz et al. (2016) and they include the derived optimal values. The range of parameter variability was chosen following the HBV model sensitivity studies reported by Osuch (2015) and Benninga (2015), It should be noted that the WHC parameter related to refreezing is very small, indicating that no refreezing process is accounted for. Following the sensitivity studies, that parameter has no influence on the model calibration, what can be explained by the lack of relevant observations. Default parameter values could be used in that
case, but that would mean that we impose information that we do not poses.

As discussed earlier, we focus on three sources of uncertainty, the first related to the HBV model input, in the form of ensemble projections of temperature and precipitation, the second, related to hydrological model parameter uncertainty and the third related to the extreme index distribution fitting uncertainty. The latter was evaluated using 10000 MC normal samples of the GEV model parameter space performed for each of behavioural parameter sets (section 4.2). As a result we
obtained 20000 daily flow simulations 130 years long for raw and bias corrected climate model projections for an ensemble of seven GCM/RCMs listed in Table 1. This gives all together 280000 flow time-series used to derive extreme flow quantiles.

We apply the QM corrected precipitation projections for the estimation of low-flow extremes and raw precipitation projections for high-flow extremes. The choice of raw precipitation data for high flow indices allowed for the elimination of errors related to bias correction in the estimates (Romanowicz et al., 2016b). On the other hand, the bias correction of precipitation projections is necessary for the low flow indices due to the drizzle effect of climate models.

## 4 Results and discussion

### 4.1 Variability of projected precipitation and temperature series

In the following section, we present an analysis of the variability of maximum precipitation and temperature series on annual basis to see the correlation between the projected climate and hydrological extremes. The idea behind presenting the precipitation changes was to show their possible relation with the changes in flow extreme indices. For a catchment of that size, annual maximum and mean sums of precipitation are well correlated with the flow patterns when the rainfall-driven flood regime prevails. The temperature changes, on the other hand, present the changes in the evaporation losses and possibly, indicate changes in the flood regime. In Fig. 2, precipitation and temperature time series for the Biala Tarnowska catchment obtained from the seven GCM/RCM models under the RCP4.5 scenario are shown. The periods cover the whole length of historical and projected years (1971-2100). The upper panel of Fig. 2 presents annual sum precipitation based on corrected precipitation projections, the annual maximum precipitation based on raw projections is shown in the middle panel, and temperature mean projections for bias-corrected data are presented in the lower panel. Low flow patterns are affected by long-term precipitation which is reflected in annual precipitation sums, whilst the high flow events have a short time scale and correspond to precipitation maxima. The results show a visible increase of the annual mean temperature and mean values of annual sums of precipitation and annual maxima do not show visible changes with time.

### 4.2 Calibration and validation of hydrological model: GLUE analysis

The calibration was performed using the observed precipitation and temperature from the Biala Tarnowska catchment and flow records from the Koszyce gauging station for the period 1971-2000 for the calibration and 2001-2010 for the validation stage. We applied the $J_{NSE}$ criterion (eq. 1) for the high flow and the $J_{logNSE}$ criterion (eq. 2) for the low flow to all the simulated flow series. The thresholds for the criteria, called likelihood thresholds were evaluated (Beven and Binley, 2014) by the "trial and error approach". As a result, two multiple sets (each including thousands of parameter sets) representing "high" and "low" flow modes of the HBV model performance have been derived. The threshold value of a goodness of fit criterion determining the GLUE-based behavioural model parameter set for high flow indices was selected at 0.55 of the $J_{NSE}$. The threshold value was selected to assure that 95% of observations lay within the 0.95 confidence

bands. The sample size of this behavioural set is 8616. The maximum Nash-Sutcliffe efficiency ($J_{NSE}$) values over the calibration and validation periods are 0.79 and 0.75, respectively. The low-flow model parameter set was selected using the *NSE* of log-transformed flow values ($J_{logNSE}$) with the threshold set at 0.3 and the sample size obtained is 1625. The maximum $J_{logNSE}$ value in calibration and validation period is 0.6212 and 0.6995 respectively. The maximum $J_{NSE}$ value
in calibration and validation is 0.7827 and 0.8128 respectively.

Fig. 3 shows the cumulative density functions (cdf) of observed daily hydrographs for the calibration and validation periods, as well as the cdf of flow estimates generated from the posterior distribution of the HBV model parameters. The upper panel presents the cdf of model predictions conditioned on the $J_{NSE}$, while the lower panel presents the cdf of predictions conditioned on the $J_{logNSE}$ criterion. Also shown are the 0.95 confidence bands in the form of dashed lines.
These confidence bands are much narrower for the $J_{logNSE}$ weights than for the $J_{NSE}$ conditioning. This indicates the strong influence of low flow predictions on the HBV model performance. Moreover, the shape of the cdfs suggests that the logarithmic transformation of flows gives a superior match of simulations to the observations in comparison with the $J_{NSE}$ criterion. In Fig. 4, the 'best' annual extreme time series of projected flow, corresponding to the deterministic 'optimal' parameter sets for the River Biala Tarnowska at Koszyce are shown. The upper panel of Fig. 4 presents annual maximum
flows, and the annual minimum flows are presented in the lower panel. These results were obtained from the HBV model simulations fed by the precipitation and temperature projections obtained from the seven GCM/RCM models under the RCP4.5 scenario for the parameter sets from the MC parameter samples giving the highest weights derived from the $J_{NSE}$ for the high flows, and $J_{logNSE}$ for the low flows, respectively. The raw precipitation projections were applied to study the high flow index whilst bias corrected precipitation data were used for the low-flow index studies. Obtained flow projections
shown in Fig. 4, follow the precipitation projections shown in Fig. 2, with annual maximum flow values even four times larger than historical events occurring after 2016 for some GCM/RCM model projections. These time series cover the whole length of the reference and projected years simulated (1971-2100) in an attempt to identify temporal variabilities in the high and low flow indices.

**4.3 Changes in extreme flow quantiles (30-year periods) due to the climate model spread**

The empirical quantiles of the future annual maximum and minimum flow projections for the 30-year periods, including the reference period 1971-2000, the near-future period 2021-2050 and the far-future period 2071-2100 are shown in Fig. 5. These results present the empirical frequency curves obtained for the best performing hydrological model parameter set for seven climate models listed in Table 1, neglecting the hydrological model parameter uncertainty. These empirical quantiles are strongly controlled by the extreme events, which may have the form of outliers (Fig. 5, left column). A
comparison of the median return periods obtained for the near- and far-future with the median in the reference period illustrates the predicted changes in quantiles. Small decreases in annual minimum flow and increases in annual maximum

flow for both near- and far-future periods can be observed. In the case of maximum annual flow (Fig. 5, left column), the reference quantile curves (dashed red lines) are always lower than those from the climate model ensemble medians (dashed blue lines), implying increases in both frequency and magnitude of annual maximum flows. Following a similar reasoning it can be deduced that the magnitudes of annual minimum flows show small decrease in the future (Fig. 5, right column).

The results for high flow extremes are consistent with those published by Osuch et al. (2016), which is not surprising when we note that the same GCM/RCM projections were used for the study catchment. The decrease of annual minimum flows in the future is also consistent with the results published by Meresa et al. (2016).

From Fig. 5, left column, upper panel, we note that the uncertainty of the empirical high-flow quantile at 30-year return period ($Q_{T30}$) related to the climate model spread exceeds 100% (600 $m^3s^{-1}$) in the near-future. In the contrary, the spread

of $Q_{T30}$ of annual maximum flows in the far-future decreases to 500 $m^3s^{-1}$ (Fig. 5, left column, lower panel). Similarly, also the low flow $Q_{T30}$ shows smaller spread for the far future period (Fig. 5, right column). The fact that the spread is more evenly distributed for minimum flows compared to maximum flows is related to the influence of the climate model spread on the simulations. It shows that climate change extremes have larger influence on flood frequency than on low flow frequency. The smaller spread of the far-future projected changes was also observed in the other climate impact

studies on the same catchment (Osuch et al., 2016) for both the RCP4.5 and RCP8.5 emission scenarios using the HBV model. Research is on-going to explain that phenomenon.

It is important to note that 30-year based quantiles are highly uncertain and show unrealistic changes which are not visible in long annual extreme series. Moreover, these changes in 30-year based quantiles are caused by inter-decadal variability and they depend on the starting year of those 30-year periods. This might explain large differences in the

estimates of future quantile changes obtained in a number of studies (Kundzewicz et al., 2017).

## 4.4 Evaluation of combined uncertainty in extreme flow quantiles for 30 and 130 year periods

The empirical frequency curves do not allow the extrapolation of a return period beyond the available number of simulation years to be performed and instead theoretical distributions fitted to the data are applied. In addition, quantiles are nonlinearly dependent on flow extremes and the averaging the best hydrological projections is not equivalent to averaging

over the whole set of realizations resulting from the behavioural parameter sets. The results of fitting the GEV distribution to annual maximum and minimum flow  for 30 year periods, including the reference period (1971-2000), the near-future period (2021-2050) and the far-future period (2071-2100) are presented in Fig. 6. The blue and light pink areas in Fig 6 present the uncertainty arising from the combined effect of the hydrological model parameter uncertainty, ensemble spread and uncertainty related to the GEV fitting, respectively for the maximum annual flow (left column) and the minimum

annual flow (right column). The quantiles of annual maximum flow show significant spread among the fitted GEV distributions, which is more pronounced for higher recurrence intervals whilst the quantiles of minimum annual flow are spread evenly. Comparison of empirical and theoretical distribution-based flood frequency curves indicates that "outliers"

(single very high flow events) have smaller influence on the distribution-based than on empirical flood frequency analyses (Figs. 5 and 6, left columns).

The uncertainties originating in the climate models and the hydrological model parameters were calculated based on a range of the differences between the 0.975 upper confidence bands and 0.025 lower confidence bands as a measure of the uncertainty in the ensemble projections that were made using multiple GCM/RCMs, hydrological model behavioral parameter sets and distribution parameter sets (FFA). When comparing the total uncertainties, it becomes clear that uncertainties from climate projections, hydrological model parameter and distribution parameter sets cannot be independently assessed to generate reliable predictive bounds for the estimates of hydrologic extremes and their characteristics.

Figure 7 presents integrated uncertainty of frequency curves of annual maximum flow (left panel) and annual minimum flow (right panel), based on the 130 years (1971-2100) of MC simulations of the HBV model. Each shading represents the contribution of a different uncertainty source (the colours are additive). The green colour denotes the hydrological model uncertainty, the blue corresponds to climate model spread and the pink colour describes the GEV distribution fit error. The red dotted lines denote the median of climate ensembles, black dotted lines denote the median of hydrological model parameter sets and the blue dotted lines denote the median related to the GEV distribution parameter fit. This kind of analysis does not illustrate the interactions between different sources of uncertainty. Generally, the uncertainty from climate models is larger than the other two for the high flow quantiles. On the other hand, for the low flow quantiles, hydrological model parameter uncertainty contributes more than the other two sources to the uncertainty of the minimum flow frequency.

The uncertainties of the quantiles of annual maximum flow due to total uncertainty accounted for (climate models, parameter sets, distribution fitting parameter sets) for the 30 year (Fig. 6) and 130 year (Fig.7) periods show significant differences. Table 3 gives a summary of confidence interval ranges obtained for the $Q_{T30}$ based on different time periods. In general, the $Q_{T30}$ estimated using the 30 year period is characterized by a much larger confidence intervals compared to the $Q_{T30}$ estimated using the 130 year long period. The differences in the width of confidence intervals vary from about 200 m$^3$s$^{-1}$ for the reference period to 1500 m$^3$s$^{-1}$ for the near future period (2021-2050) compared to the 130 year period $Q_{T30}$ estimates. Due to the extrapolation errors, that difference will increase substantially for the $Q_{T100}$ index, thus questioning the usefulness of those estimates. The relative differences obtained for the annual minimum flow $Q_{T30}$ estimates are smaller, suggesting that low flow quantiles are less susceptible to the errors related to the length of the evaluation period.

The comparison of flood frequency curves obtained from 30 and 130 year series of annual extremes is not possible. However what we compare is the uncertainty range of flow quantiles derived from 30 year and 130 year-long annual extreme series. Such comparison is not related to non-stationary condition of the extremes but rather to the number of events included in the frequency derivation.

The results of the study show that the uncertainties in extreme maximum and extreme minimum indices behave differently. In extreme high flow, larger uncertainty is observed from the climate model (ensemble) spread than from the other sources. In contrast, for low flows the uncertainty related to hydrological model parameters has a larger impact than the other uncertainty sources studied. The important role of hydrological model uncertainty in low flow predictions has already been noticed in forecasting (Beninga et al., 2017). That effect can be explained by the ratio of the prediction noise (in this case described by the hydrological model uncertainty) to the input signal which is much higher for low flows. Demirel et al. (2013b) explored the influence of uncertainty in input, hydrological model parameters and initial conditions on a 10-day ensemble flow forecasts. The results showed that parameter uncertainty had the largest effect on the mean value low flow forecasts, which is consistent with the present paper findings. This implies that ignoring one of the three uncertainty sources may cause great risk to future hydrological extreme adaptations and water resource planning and management. Steinschneider et al. (2012) used the formal statistical approach to quantify uncertainty quantiles of monthly flow projections including climate, hydrological model parameter and distribution fit uncertainties. In this study we applied the non-formal statistical approach for projections of daily annual extreme low and high flow indices.

### 4.5 Variance decomposition of quantile $Q_{T30}$ values

Fig. 8 shows the results of an application of the ANOVA variance decomposition technique to the percentage change of $Q_{T30}$ quantiles derived for the near-future period 2012-2050 relative to the reference period 1971-2000 for high flows (left panel) and low flows (right panel). The analysis was performed on the flow simulation sets including all three sources of uncertainty and conditioned by the $J_{NSE}$ weights for high flow quantiles and $J_{logNSE}$ weights for low flow quantiles. The symbols correspond to those used in Eq. 3. The correlation between parameters is marked with a star.

The sensitivity analysis presented in Fig. 8 confirms our earlier results on the major influence of the climate model spread on the total $Q_{T30}$ variability for high flows and supreme influence of hydrological model parameters on the variability of low flow Q$Q_{T30}$. There is also a difference in the influence of distribution fit uncertainty, which is much larger for low flow $Q_{T30}$ variability than for high flow. The sensitivity analysis also confirms the inter-dependence of different sources of uncertainty, visible mainly for high-flow extremes.

### 5. Conclusions

The results of the research on the assessment of the uncertainty of extreme hydrological indices can be summarized in the following points:

(i)    In order to eliminate influence of bias correction on flow maxima, the analysis of changes in the quantiles of maximum annual flow projections was based on the raw projections of precipitation. However, the analysis

of low flow projections was based on the bias-corrected data to avoid the drizzle effect which affects the low flow characteristics.

(ii)    Conditioning of the hydrological model was performed using different criteria for low and high flows in order to ensure the best model fit for the extremes. This does not solve the problem of the non-stationarity of model parameters but permits a focus on parameter sets adequate for low and high flow regimes.

(iii)    Analysis of the influence of the length of time series records on the uncertainty bands of the low and high flow quantile estimates and their changes suggests that the range of quantiles of return periods $Q_{T30}$ are up to four times smaller when the long-term flow projections are used. The low flow $Q_{T30}$ quantiles are less influenced by the length of the record.

(iv)    Taking into account the three uncertainty sources considered, the uncertainty of the estimate of 1-in-100 year return maximum flow based on the 1971-2100 time series exceeds 200% of its median value with the largest influence of the climate model uncertainty; whilst the uncertainty of the 1-in-100 year return minimum flow is of the same order (i.e. exceeds 200%), but it is mainly influenced by the hydrological model parameter uncertainty.

(v)    A sensitivity analysis using ANOVA performed on the relative total uncertainty for $Q_{T30}$ quantiles shows the largest influence of climate model and interactions between climate model and distribution fit uncertainty for high flows, whilst uncertainty of hydrological model parameters and distribution fit have the largest influence on the uncertainty of low flow quantiles.

(vi)    The analyses were performed for a catchment with stationary future extreme flow projections; in the case of nonstationary extreme flows, nonstationary frequency analysis would have to be applied with even larger uncertainty of extreme estimates than those presented here.

(vii)    The study has pointed to the need to explore different approaches to projections of climate change.

**Acknowledgements**. This work was supported by the project CHIHE (Climate Change Impact on Hydrological Extremes), carried out in the Institute of Geophysics Polish Academy of Sciences, funded by Norway Grants (contract No. Pol-Nor/196243/80/2013) and partly supported within statutory activities No 3841/E-41/S/2017 of the Ministry of Science and Higher Education of Poland. The hydro-climate data were provided by the Institute of Meteorology and Water Management (IMGW), Poland.

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

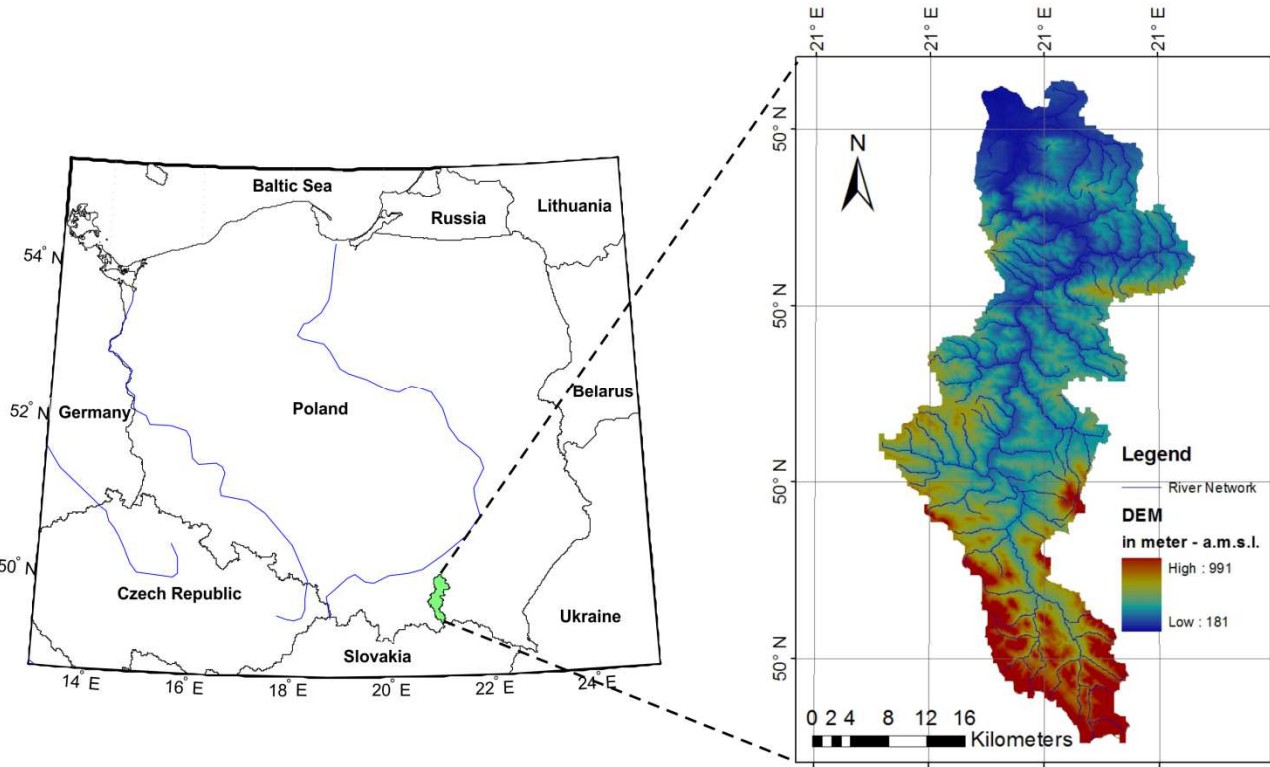

5   **Figure 1.** The location of the study catchment.

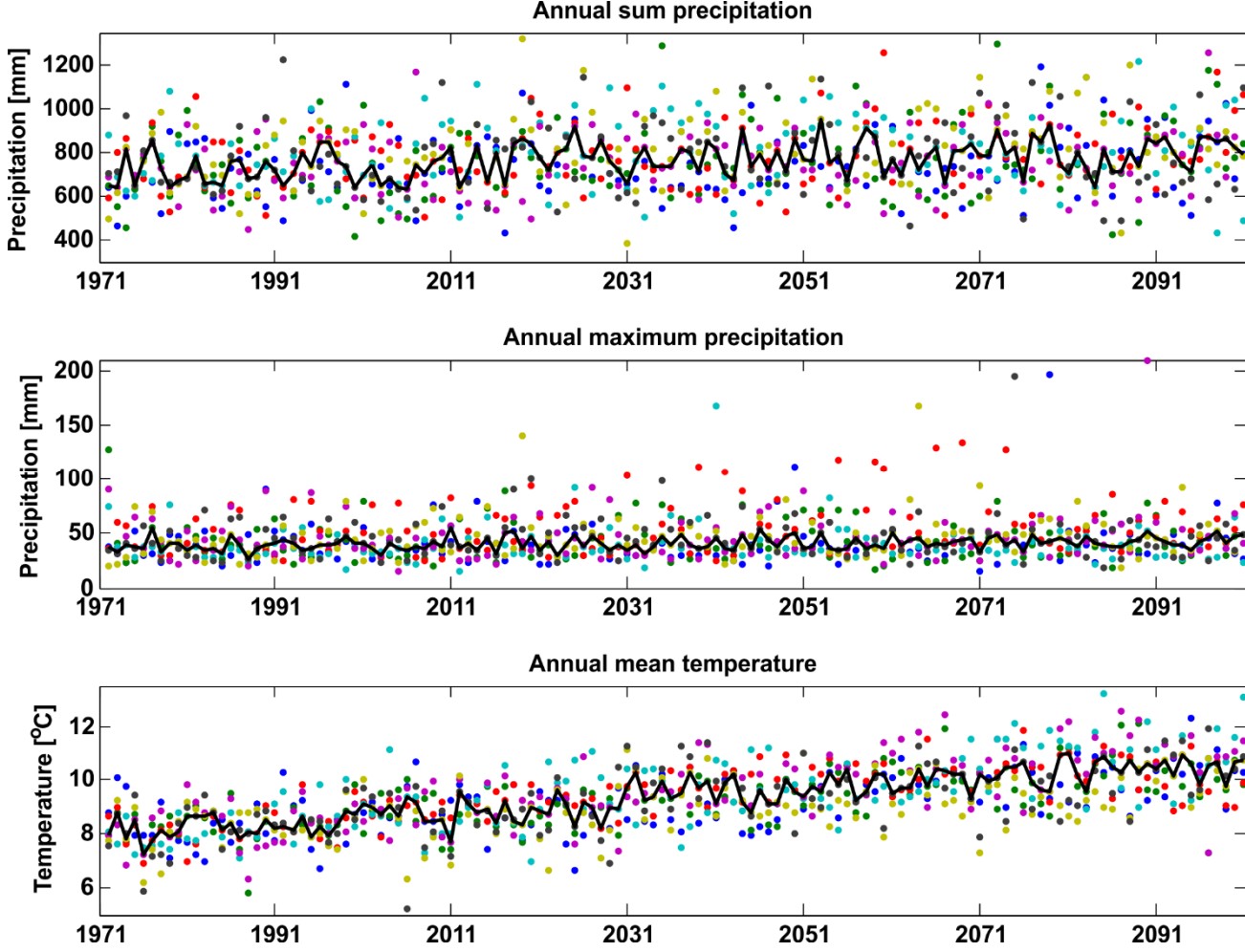

Figure 2. Climate model projections for the Biala Tarnowska catchment in the 1971-2100 period based on the seven climate models from the GCMs/RCMs combinations; upper panel: projected bias-corrected annual sum precipitation; middle panel: projected raw annual maximum daily precipitation; lower panel: projected bias-corrected annual maximum daily temperature; each colored dot represents an individual climate model and black line plot represents the median of the seven GCM/RCM combinations.

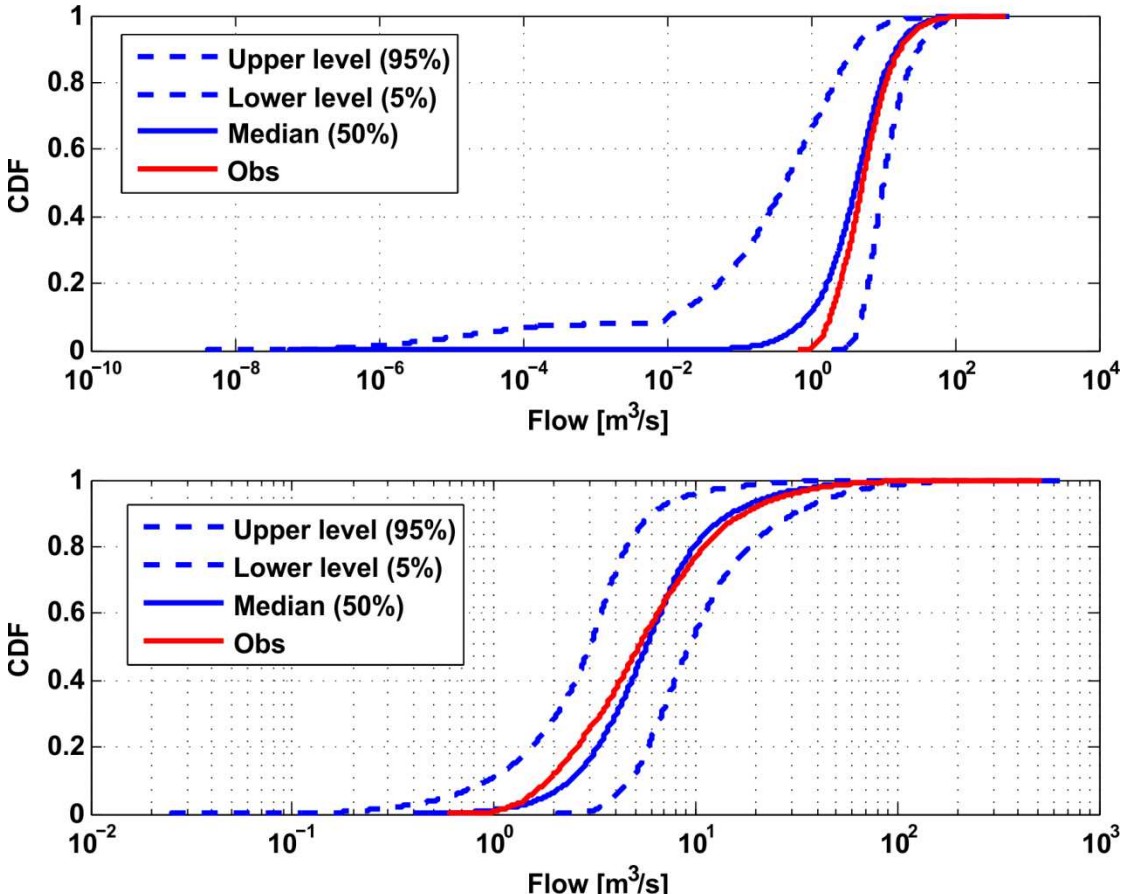

**Figure 3**. The cdf of flow for the calibration period for the HBV model; the upper panel presents model predictions conditioned on the $J_{NSE}$, while the lower panel presents the predictions conditioned on the $J_{logNSE}$ criterion; the cdf of observations (red line) are shown against the cdf of the HBV predictions (blue line) and the associated 95% confidence bounds (dashed line).

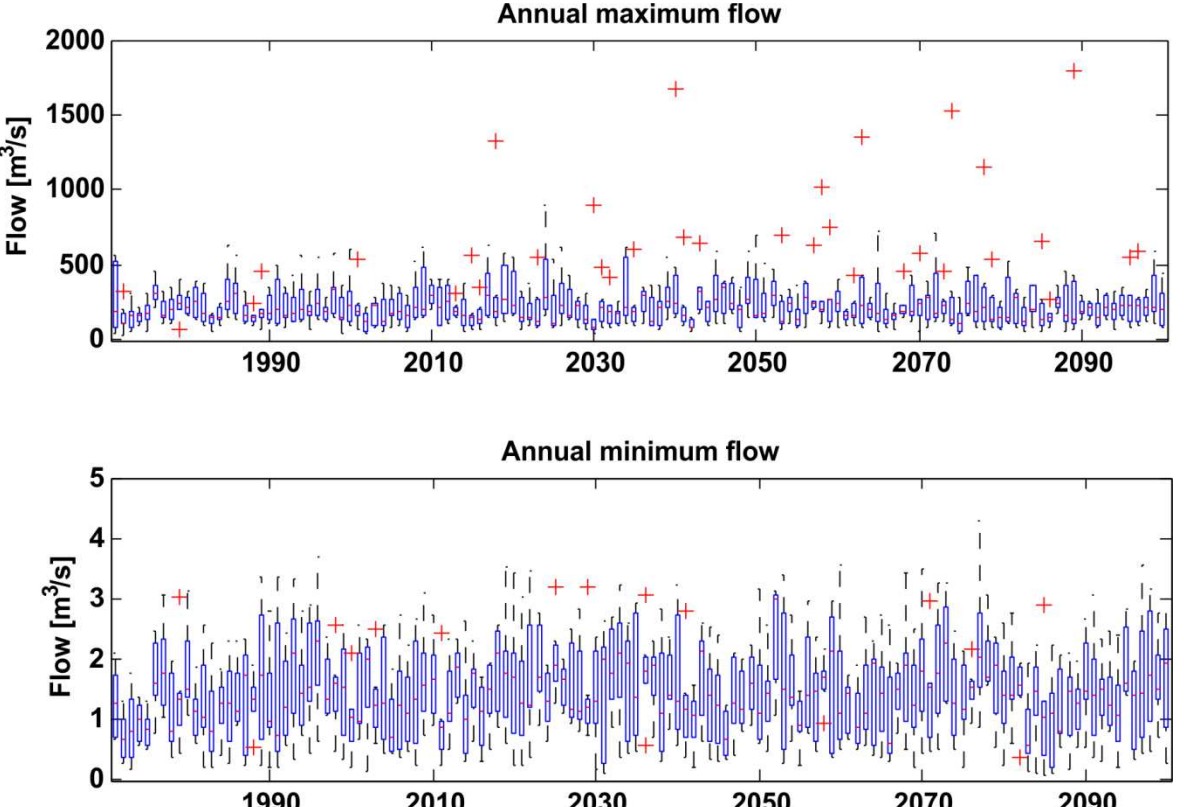

**Figure 4**. The HBV model extreme projections for the best HBV model parameter sets for the Biala Tarnowska at Koszyce in 1971-2100 based on seven climate models from the GCMs/RCMs ensemble; upper panel: projected annual maximum daily flow for the HBV parameter set corresponding to the best $J_{NSE}$ value; lower panel: projected annual minimum daily flow for the HBV parameter set corresponding to the best $J_{logNSE}$ value; red dashed line shows an ensemble mean for the 1971-2100 period.

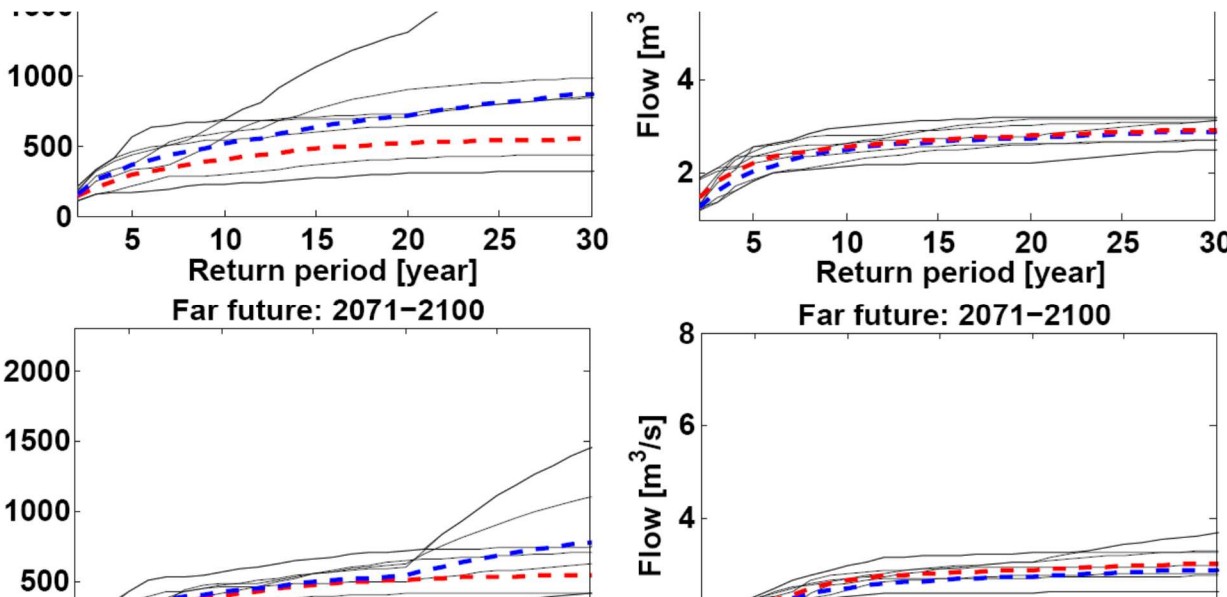

**Figure 5**. Empirical flow quantiles of annual maximum flow (left column) and annual minimum flow (right column) and future climates (near and far future) for the best sets of the HBV model parameters and seven GCM/RCM model realizations; green dashed line denotes the mean value from all the GCM/RCM model realizations in each period (near and far future period), red dashed line denotes the averaged results obtained for the reference period; each black line represents individual climate model.

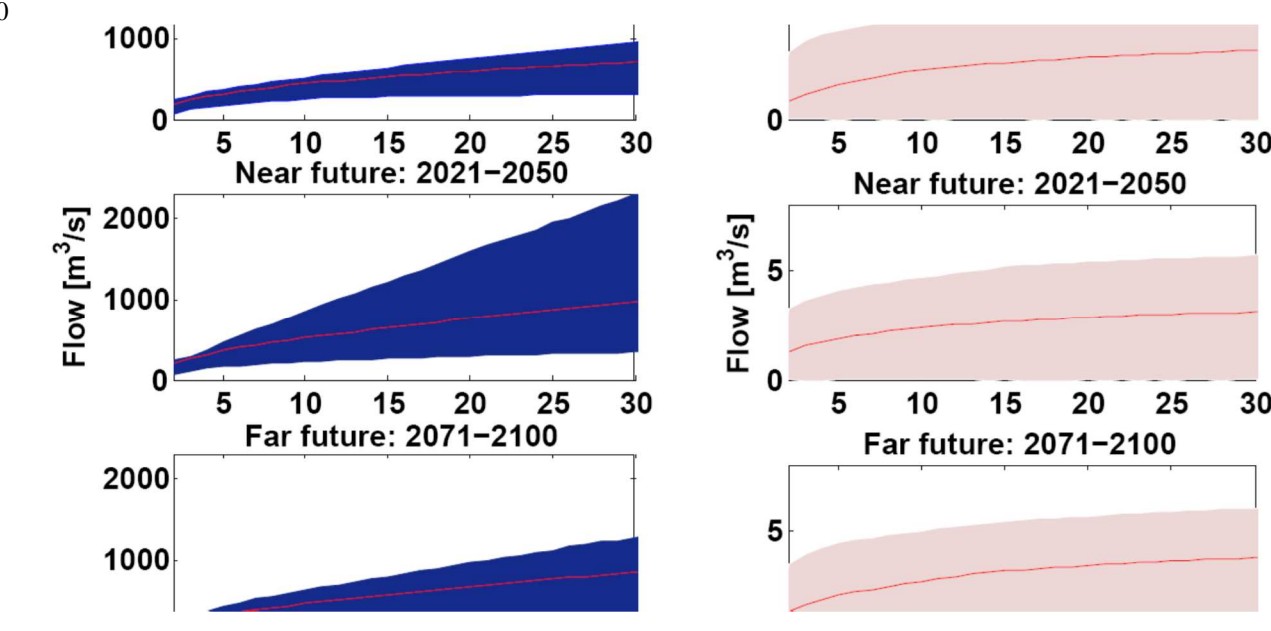

**Figure 6**. Total uncertainty ranges of annual extreme flow quantiles based on the GEV distribution for projections over 30-year periods for the Biala Tarnowska at Koszyce; the left column presents the annual maximum flow, the right column shows annual minimum flow; upper panels - the reference period (1971-2000); middle panels - near future (2021-2050); lower panels - far future (2071-2100) periods.

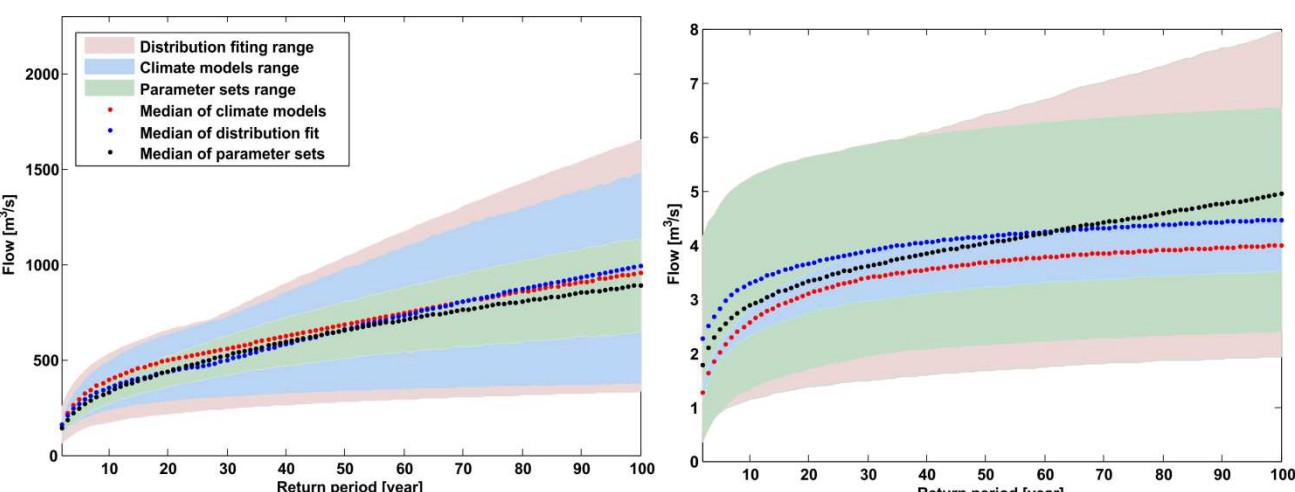

30 **Figure 7.** Total uncertainty ranges of flow quantiles for the River Biala Tarnowska at Koszyce based on the GEV distribution for projections over 130-year period (1971-2100); the left panel presents annual maximum flow and the right panel presents annual minimum flow; the blue shaded area denotes the climate model uncertainty, the green shaded area denotes the hydrological model

uncertainty and the pink shaded area denotes the distribution fit uncertainty; red dotted lines denote the median of climate ensembles, black dotted lines denote the median of hydrological model parameter sets and blue dotted lines denote the median of the distribution fit.

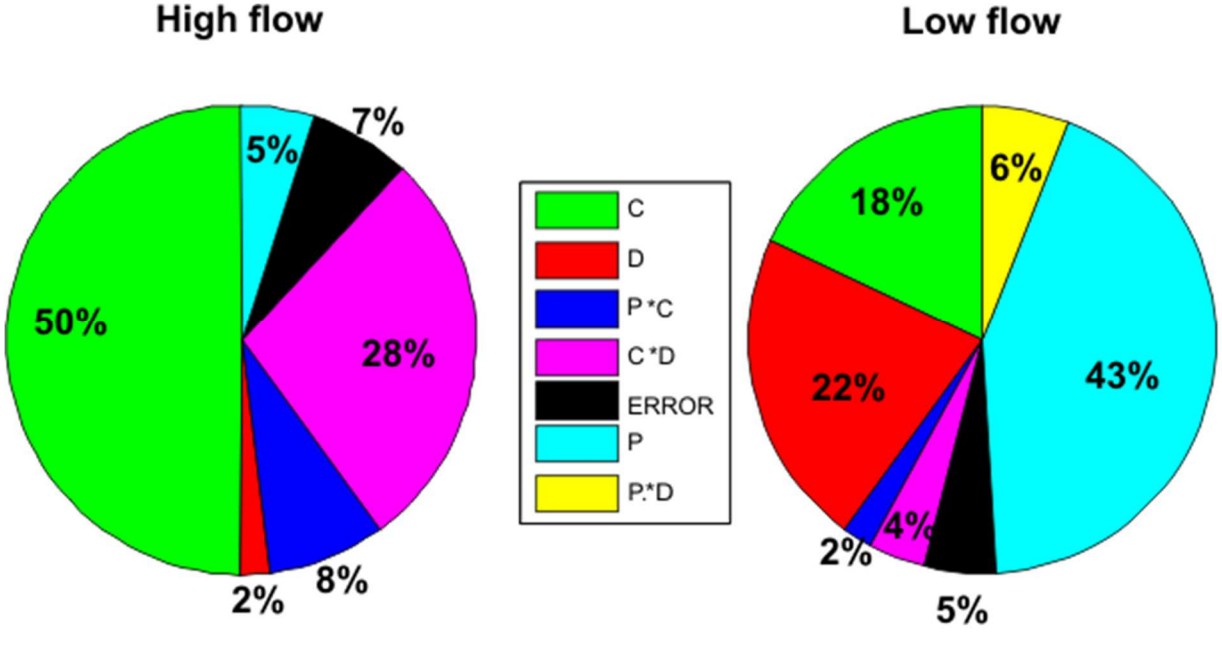

**Figure 8**. Total variance in estimates for the percentage change in $Q_{T30}$ in 2021-2050 relative to the 1971-2000 reference period. Each
10   colour represents the relative contribution of uncertainty in percent; *C* denotes climate model; *D* – distribution fit; *P* – hydrological model parameters; *ERROR* denotes the Gaussian error (Eq. 3); a "star" denotes the correlation between the factors (*C*, *D* and *P*).

| GCM | RCM | expansion name | Institute |
|---|---|---|---|
| EC-EARTH | RCA4 | Regional-scale model | Swedish Meteorological and Hydrological Institute |
| EC-EARTH | HIRHAM5 | Atmospheric model | Danish Meteorological Institute |
| EC-EARTH | CCLM-4-8-17 | Community land model | NCAR UCAR |
| EC-EARTH | RACMO22E | Regional atmospheric climate model | Meteorological institute |
| MPI-ESM-LR | CCLM4-8-17 | Community land model | Max Planck Institute for Meteorology |
| MPI-ESM-LR | RCA4 | Regional-scale model | Max Planck Institute for Meteorology |
| CNRM-CM5 | CCLM4-8-17 | Community land model | CERFACS, France |

Table 1. List of RCM/GCMs models used in this study

Table 2. HBV parameter ranges: upper band (UB), lower band (LB), unit; fixed parameters have lower and upper bands equal.

| Parameter | description | LB | UB | Unit |
|---|---|---|---|---|
| FC | maximum soil storage | 0.1 | 250 | mm |
| BETA | Shape coefficient | 0.01 | 7 | - |
| LP | SM threshold for reduction of evaporation | 0.1 | 1 | - |
| KS | recession coefficient for runoff from base flow | 0.0005 | 0.3 | $d^{-1}$ |
| PERC | percolation rate occurring when water is available | 0.01 | 100 | $mm\ d^{-1}$ |
| CFMAX | snowmelt rate | 0 | 20 | $mm\ ^oC^{-1}\ d^{-1}$ |
| DTTM | TT correction to give a threshold temperature | 0.1484 | 0.1484 | $^oC$ |
| CFR | refreezing factor | 0.2779 | 0.2779 | - |
| WHC | water holding capacity of snow | 0.001 | 0.001 | $mm\ mm^{-1}$ |
| ALFA | measure for non-linearity of flow in quick runoff | 0.2255 | 0.2255 | - |
| KF | recession coefficient for runoff from quick runoff | 0.2826 | 0.2826 | $d^{-1}$ |
| CFLUX | rate of capillary rise | 1.0003 | 1.0003 | $mm\ d^{-1}$ |
| TTI | temperature threshold interval length | 2.5 | 2.5 | $^oC$ |
| TT | temperature threshold for snowfall | 1.0145 | 1.0145 | $^oC$ |

Table 3. Change in width of 0.95 confidence intervals for $Q_{T30}$ for annual maximum and minimum flow estimated using time periods of a different length (30-year and 130-year- long).

| Evaluation period | 1971-2000 | 2021-2050 | 2071-2100 | 1971-2100 |
|---|---|---|---|---|
| Max flow($\Delta Q_{T30}$) [m³/s] | 640.5 | 1942.6 | 898.9 | 459.4 |
| Min flow($\Delta Q_{T30}$) [m³/s] | 4.7 | 5.0 | 5.2 | 4.4 |

