# Peer review of "The critical role of uncertainty in projections of hydrological extremes"

_Hydrology and Earth System Sciences, 2016_

## Referee Comment (RC1) · Anonymous Referee #1 · 21 Dec 2016

General comments

This paper is about the uncertainty of extreme flows with climate change. For that purpose, the authors use seven combinations of global climate models (GCMs) and regional climate models (RCMs) with one greenhouse gas concentration scenario to represent uncertainty in climate change. Furthermore, they use the GLUE method to represent hydrological parameter uncertainty and uncertainty in extreme value distribution parameters to represent the uncertainty in the statistical extreme value distribution. These three sources of uncertainty are investigated using the HBV hydrological model applied to a medium-sized Polish catchment.

Although the topic is interesting and relevant for this journal, the paper is moderately

written, lacks clarity in parts of the methodology and only briefly discusses results and insufficiently puts outcomes into perspective. For instance, the seemingly arbitrary choice to consider the three uncertainty sources is not justified. Are these three sources the most important ones or the easiest ones to quantify? Furthermore, the uncertainty due to the use of a particular extreme value distribution is not clearly and completely incorporated. A final example is the presentation and analysis of results, such as the analysis of annual maximum precipitation and temperature in relation with annual maximum flows and in particular annual minimum flows. In this case and several other cases it is not always clear which results are shown, why they are shown and what can be concluded from the results. Many other specific (and important) comments can be found below. Furthermore, the English writing style and grammar is moderate (including several typos); some examples can be found in the section 'technical corrections'.

Specific comments

1. P1, L7-9: It is not clear what is meant with a 'multi-model approach' and which steps are followed.

2. P2, L9-11: The first question probably is related to the magnitude of the uncertainty, since this is still largely unknown and not systematically investigated.

3. P2, L15-16: ". . . can never be accurately evaluated . . ." is a very strong statement, please rephrase.

4. P2, L24-P3, L2: The authors mainly consider hydrological model and parameter uncertainty in their review. It might be worthwhile to firstly give an overview of all uncertainties involved in this type of studies including a classification. One such classification could be input, (hydrological) model system and output, and the literature can be reviewed accordingly. Now, uncertainties in the input (scenarios, GCMs, RCMs, downscaling, initial conditions etc.) are hardly reviewed. A complete overview of the uncertainties will also enable a better justification of the uncertainty sources considered

in this study (see also page 3, lines 4-5).

5. P3, L14-15: The question is whether you can determine the uncertainty due to the choice of the extreme value distribution ('distribution fit') using time series of different lengths. When assessing effects of time series with different lengths on the results you might get an estimate of the influence of data quantity on the uncertainty in the results, but not of the influence of the goodness-of-fit of the distribution on the uncertainty. Furthermore, it seems only part of the statistical uncertainty is assessed in this way, since for instance the influence of different extreme value distributions and extrapolation uncertainty is not taken into account.

6. P3, L29-30: How many precipitation stations have been used to assess the catchment average precipitation (assuming lumped hydrological modelling has been carried out)? Has any elevation (or other) correction been incorporated?

7. P4, L11: An important uncertainty source in climate impact studies is the uncertainty due to greenhouse gas emission scenarios. Hence, a limitation of this study is the use of only one emission scenario (RCP4.5) while one would expect the use of at least two scenarios (which are available in EURO-CORDEX). At least the authors should explain the implications of this limitation for their results.

8. P5, L9: Why is QM applied in this study? The reasoning behind this choice is not completely clear from the preceding sentences.

9. P5, L18-19: Did Osuch et al. (2015) model the same catchment as in this study and therefore, can it be assumed that the same five parameters are sensitive? And are the same five parameters sensitive for low flows and for high flows? That would be remarkable.

10. P6, L15-16: How many Monte Carlo simulations have been executed and is this number sufficient (compare with literature)?

11. P6, L22: Is it common practise to determine the thresholds in an iterative way? The

determination of the threshold based on the requirement that 95% of the observations should be in the 95% confidence interval seems to be reasonable. However, please refer to other studies employing the same approach.

12. P7, L4-5: In general it is doubtful whether distributions with a 'large' number of parameters will model data in a more accurate way than distributions with a small number. This partly depends on the data quantity and quality and similarly as in hydrological modelling there will be a balance between the complexity of the distribution (i.e. number of parameters) and the amount of data (and quality).

13. P7, L7-8: What does an 'overall good performance' mean? Compared to which other distributions?

14. P7, L25-27: It is not completely clear why the analyses are performed for a period of 130 years. Since the manuscript is about impacts of climate change on hydrological extremes, you would expect a comparison between historic and future climate conditions. Furthermore, climate change automatically implies the existence of non-stationarity and as such, by considering a period of 130 years assuming stationarity by using the same extreme value distribution will result in serious flaws.

15. P8, L7-12: The idea behind this section is not clear. Why is the trend in daily annual maximum precipitation and temperature analysed while the interest is in uncertainty in hydrological indices with climate change? Moreover, why is the daily annual maximum precipitation of interest and not for instance the two-day or three day precipitation (which might be stronger correlated to annual maximum discharge values)? Which temporal resolutions of precipitation are relevant for annual minimum flows? And what is the supposed role of daily annual maximum temperature values?

16. P8, L14-20: How have the different criteria for high and low flows been applied in continuous hydrological modelling for periods of 30 years (or more)? When is the 'high flow' parameter set being used and when the 'low flow' one? What is the threshold for low flows and high flows; a specific discharge value or exceedance frequency?

17. P9, L5: Which best parameter sets are meant here? When is the best low flow parameter set used and when the best high flow parameter set?

18. P9, L7-8: 'twice as large'; where do we see that?

19. P9, L12-22: This evaluation is not clear to me. Why do the authors evaluate results at a monthly scale? How can you assess annual maximum flows for each month? What do the authors mean with 'range' of annual maximum flows?

20. P10, L9-10: The decrease in the spread of Q30 in the far future compared to the near future is strange. The authors should reflect on this. Is it related to the fact that only one RCP scenario is taken into account?

21. P10, L20-22: Also this observation needs discussion. Why the spread is more evenly distributed for minimum flows compared to maximum flows?

22. P11, L13-14: Are the relative differences for annual minimum flows also smaller?

23. P12, L7-9: This is an interesting topic, but has not been investigated in this study since only one catchment has been considered.

24. P12, L11-14: This is an interesting result assuming that all methodological steps are logical and correctly carried out. What is the reason for the importance of uncertainty due to climate models for high flow and the important of hydrological model parameter uncertainty for low flows? This is very important and interesting to discuss.

25. P12, L23-24: What do the authors mean with 'this allows the problem of nonstationarity of model parameters to be avoided'?

26. P12, L29-31: This statement seems to be obvious; the larger the ratio of return period vs. data length the higher the uncertainty. However, this extrapolation uncertainty is not explicitly assessed in this manuscript.

27. P23, Table 2: The ranges defined by the lower and upper bounds frequently do not match with the optimal values (e.g. for ALFA, PERC, CLFUX). Can you explain this?

Furthermore, some lower and upper bounds are exactly the same. Does this indicate that these parameters are deterministic? What about CFMAX (not mentioned as sensitive in section 3.3)? Finally, an upper bound of 2 for LP is impossible and an optimum value of 1 is remarkable at least (it would mean only potential evapotranspiration under fully saturated conditions).

Technical corrections

1. P1, L11: What is the distribution fit?

2. P1, L13: What kind of weighting do the authors mean?

3. P1, L16: What is the difference between climate model variability and climate projection ensemble spread? Please use a consistent terminology.

4. P2, L3: What is inverse modelling in this respect? Is this term commonly used for calibration and validation purposes based on observed (historic) data?

5. P2, L6: "weighting" instead of "weighing".

6. P3, L8: What is the 'relevant variability' of extreme index estimates?

7. P3, L19: The case study has already been mentioned.

8. P3, L30: The maximum daily precipitation? During which period?

9. P3, L30-31: Which period for the streamflow) Isn't 0.4 m3/s a very low value for catchment area of about 1000 km2?

10. P4, L12-14: Why do the authors use these complex abbreviations for the GCM-RCM combinations? It is not clear what the meaning of all the numbers is. Try to be consistent with the descriptions in Table 1.

11. P5, L12: Do you have a reference for the Matlab version of HBV?

12. P5, L15-17: Only 12 out of 14 HBV parameters are mentioned. In which routines can we find CFLUX and PERC (see line 19)?

13. P5, L17: 'routines' instead of 'routing stage'?

14. P6, L24-P7, L3: This general description of the GEV distribution is not necessary here and can be found in many text books.

15. P7, L16-17: What do the authors mean with "... aggregated speared of flow quantile change ..."?

16. P7, L19: 'squared' instead of 'squere'.

17. P7, L22: The title suggests that the results of this study will be described. Please rephrase the title.

18. P7, L23: Different temporal resolutions? Shouldn't it be different lengths of data periods?

19. P7, L18: The meaning of all variables should be explained in the text.

20. P8, L6: "Results and discussion"?

21. P9, L2: 'the 10-year moving average from the ensemble mean'?

22. P9, L15-16: Fig. 5a is mentioned twice.

23. P9, L29-30: Decreases in minimum flows and increases in maximum flows? Shouldn't it be the other way around (according to the caption of Fig. 6)?

24. P10, L6-7: Here, the annual minimum flows increase (see previous comment).

25. P10, L9: What is Q30? Commonly, that is a discharge with a non-exceedance frequency of 30%. However, here it seems to be an annual maximum flow with a return period of 30 years?

26. P11, L7: 'Table 4' instead of 'Table 3'.

27. P11, L26-P12, L2: The first part of the conclusions can be omitted (can be part of introduction section).

28. P12, L9: 'hydrological parameter uncertainties' instead of 'hydrological model uncertainties'?

29. P12, L24-27: This is a repetition of lines 11-14.

30. P13, L3: A paper in preparation should not be included in the reference list.

31. P13-17: The reference list and referencing contain many errors, typos and inconsistencies. This should be carefully and thoroughly double-checked.

32. P18, Fig. 1: What is the unit of the DEM map?

33. P18, Fig. 2: The interquantile range of what? Of the seven GCM-RCM combinations? In that case it would be better to show the individual model results, i.e. one annual maximum for each combination so 7 points per year.

34. P19, Fig. 3: In particular the scale of the upper panel looks strange. Flows in cubic mm? How accurate is your model? Please use the same (realistic) x-axis ranges.

35. P19, Fig. 4: This figure (and also Fig. 2) is too small. What do we see here?

36. P20: The differences between historic and future periods cannot be clearly seen in these figures.

37. P21, Fig. 6: What are the different lines in these figures? And is baseline and reference period the same?

38. P21, Fig. 7: In the caption 'right hand panel' is mentioned twice.

39. P22, Fig. 8: Idem, annual minimum flow is mentioned twice.

40. P23, Table 1: Which meteorological institute is connected to RACMO?

41. P23, Table 2: The caption is not clear.

42. P24, Table 4: What do the authors mean with 'change in width of ...'? What compared to what?

---

## Referee Comment (RC2) · Anonymous Referee #2 · 15 Jan 2017

The authors assess the effect of different uncertainty sources on climate change projections. The presentation of the results is easy to follow and interpret. Especially Figure 9 is very informative. However, there is room for improvement using specific comments and checklist below. I recommend major revision as the model calibration part is not clear.

Specific Comments: 1. Table 2: Optimal values of some parameters are out of lower and upper limits e.g. CFMAX which cannot be reached by an algorithm e.g. SCEUA, CMAES etc. How was this achieved by a calibration algorithm? Did you follow a manual calibration scheme? 2. Demirel et al (2013a) is in the reference list but not in the text. 3. Please explain the abbreviations used at legend in figure caption. The legend of Fig

8 is confusing: "distn"? 4. Did you compare uncertainty in HBV model parameters with other studies (Addor et al., 2014; Demirel et al., 2013b; Osuch et al., 2015) using HBV model for forecasting hydrological extremes? How would the results overlap for 10 day forecast (Demirel et al., 2013b) and long term climate predictions in EUROCORDEX (dataset used in this study)? 5. Fig5: Parameter uncertainty should be presented differently to assess the contribution of each parameter uncertainty to total uncertainty. From this figure the reader can't see the most uncertain parameter. A figure similar to Figure 4 in Demirel et al (2013b) or Fig9 in the current manuscript can be very useful for modelers. This can be easily done as the GLUE results would allow such ranking. 6. Conclusion 2 (ii): Please explain the drizzle effect? Not clear. 7. Section 3.6 and Conclusion 5 (v): Is ANOVA method a global or local sensitivity analysis method? Can interactions (parameter etc) be assessed using this method? Why ANOVA is used instead of other elementary and global methods e.g. Morris, SOBOL, PEST, FAST etc. These aspects of the ANOVA method should be described in section 3.6 and conclusions should follow these details. 8. Conclusion bullets are confusing. Two times "iv" exists and sentences are not clear. There are typos too. For example Conclusion vi should start with capital. Please rephrase them with short and clear conclusions. And relate them to the results section. Bullet conclusions in Demirel et al (2013b) can be an example. For each result section one paragraph is given in conclusion.   HESS REVIEW CHECK-LIST 1. Does the paper address relevant scientific questions within the scope of HESS? Yes, the authors assess different uncertainty sources in climate projections. They found out that the model parameters can be more uncertain than some other important sources i.e. the climate model variability and distribution fit uncertainty for the lowflow extremes. This outcome is in line with the findings of Demirel et al 2013, WRR (DOI: 10.1002/wrcr.20294). 2. Does the paper present novel concepts, ideas, tools, or data?

Yes, this is an extensive analysis of uncertainties on low and high flows. 3. Are substantial conclusions reached?

Yes, especially ranking of uncertainties is significant. 4. Are the scientific methods and assumptions valid and clearly outlined?

Yes, the authors explained the methods clearly. 5. Are the results sufficient to support the interpretations and conclusions?

Yes 6. Is the description of experiments and calculations sufficiently complete and precise to allow their reproduction by fellow scientists (traceability of results)?

This is a research paper, a case study from Poland. 7. Do the authors give proper credit to related work and clearly indicate their own new/original contribution?

Yes 8. Does the title clearly reflect the contents of the paper?

Yes 9. Does the abstract provide a concise and complete summary?

Yes 10. Is the overall presentation well structured and clear?

Yes but sub-titles in methods and results sections should be consistent. Are the methods for section 4.7 explained in section 3.6? What do you mean by seasonal flows? Seasonal low and/or high flows? 11. Is the language fluent and precise?

Yes 12. Are mathematical formulae, symbols, abbreviations, and units correctly defined and used?

Yes 13. Should any parts of the paper (text, formulae, figures, tables) be clarified, reduced, combined, or eliminated?

No 14. Are the number and quality of references appropriate?

Yes, enough 15. Is the amount and quality of supplementary material appropriate?

N/A

---

## Author Comment (AC1) · 9 Mar 2017

Dear Editor and Reviewers,

Thank you very much for the constructive comments that will help to considerably improve and clarify the manuscript. The reviewers, put enormous effort into proof-reading our paper line by line and trying to clarify all the less-than-satisfactory statements and mistakes. We believe that the input will improve the manuscript significantly. All comments have been addressed point-by-point. Following the reviewers' feedback we will make the corresponding changes in the manuscript. Reviewers' comments are in italics.

Anonymous Referee #1

RC1. General comments

This paper is about the uncertainty of extreme flows with climate change. For that purpose, the authors use seven combinations of global climate models (GCMs) and regional climate models (RCMs) with one greenhouse gas concentration scenario to represent uncertainty in climate change. Furthermore, they use the GLUE method to represent hydrological parameter uncertainty and uncertainty in extreme value distribution parameters to represent the uncertainty in the statistical extreme value distribution. These three sources of uncertainty are investigated using the HBV hydrological model applied to a medium-sized Polish catchment. Although the topic is interesting and relevant for this journal, the paper is moderately written, lacks clarity in parts of the methodology and only briefly discusses results and insufficiently puts outcomes into perspective. For instance, the seemingly arbitrary choice to consider the three uncertainty sources is not justified. Are these three sources the most important ones or the easiest ones to quantify? Furthermore, the uncertainty due to the use of a particular extreme value distribution is not clearly and completely incorporated. A final example is the presentation and analysis of results, such as the analysis of annual maximum precipitation and temperature in relation with annual maximum flows and in particular annual minimum flows. In this case and several other cases it is not always clear which results are shown, why they are shown and what can be concluded from the results. Many other specific (and important) comments can be found below. Furthermore, the English writing style and grammar is moderate (including several typos); some examples can be found in the section 'technical corrections'.

AC1. General answer The paper lacks clarity in parts of the methodology and only briefly discusses results and insufficiently puts outcomes into perspective. Following the reviewer's general and specific comments, the clarity of the methodology will be

improved and the outcomes will be described in a wider perspective.

For instance, the seemingly arbitrary choice to consider the three uncertainty sources is not justified. Are these three sources the most important ones or the easiest ones to quantify

The choice of three particular sources of uncertainty, namely, a set of climate model ensembles, hydrological model parameter uncertainty and uncertainty in fitting extreme value distribution, was dictated by one of the aims of the research – i.e. an assessment of influence of hydrological model uncertainty on projections of low- and high-flow extremes and the relative contribution of that "predictive" uncertainty in the spread of extreme indices related to the climatic model spread and the distribution fitting error.

This choice followed from a discussion on all the sources of uncertainties and a review of research done so far on the assessment of uncertainty of projections of hydrological extremes. The "predictive" model uncertainty is the only one which can be decreased when conditioned on the observations. The other sources of uncertainty have an "epistemic" nature and cannot be decreased. Bearing in mind the aims of the study, we restricted the sources of epistemic uncertainty to those which have the largest impact – i.e. climate model spread, omitting the uncertainty related to bias correction or geography. The error related to the distribution fit was included as an essential part of the extreme index evaluation, which requires extrapolation of annual maximum or minimum flow distributions to higher order quantiles (e.g. 1-in-100 year, or 1-in-200 year). In this paper, the error related to the evaluation of maximum and minimum annual flow statistic was treated as epistemic, that means, not conditioned on real observations.

In addition, Osuch et al. (2016) presented the influence of emission scenario, climate model, bias correction method and geography on flow indices in a case study that included the same catchment, Biala Tarnowska. Therefore we wanted to avoid the repetition.

In this regard, our paper is an extension of the former paper, focusing on the influence

of hydrologic model uncertainty on annual maximum and minimum flow projections. In our opinion, including the other sources of uncertainty would obscure our aim.

The uncertainty due to the use of a particular extreme value distribution is not clearly and completely incorporated.

The choice of extreme value distribution followed the validation of suitability of this distribution to describe the projected annual maximum and minimum flows using the probability plots. The MATLAB- based GEV distribution fitting algorithm was applied to all the climate models and the a posteriori hydrological model parameter set. This algorithm provides the estimates of 0.95 confidence bands for the distribution parameters. These parameters were subsequently used to obtain upper and lower confidence bands of the distribution through the inverse GEV model. In order to simplify the procedure, instead of sampling from the GEV parameters within the parameter space common to all hydrologic and climate model simulations, we sampled from each set of parameters assuming a normal distribution with the variance specified by the parameter upper and lower 0.95 confidence value, and in addition, assuming the independence of the GEV model parameters. The obtained 0.95 GEV distribution confidence values were used to estimate the spread of results related to the distribution fit.

Bearing in mind the large number of simulations, it was not possible to choose the best distribution for each projected time series. Furthermore, the aim of this study is to assess the ranges of uncertainty of extreme indices rather than their exact values.

A final example is the presentation and analysis of results, such as the analysis of annual maximum precipitation and temperature in relation with annual maximum flows and in particular annual minimum flows. In this case and several other cases it is not always clear which results are shown, why they are shown and what can be concluded from the results.

We agree with the reviewer that the presentation should be much improved and clarified. The following explanation will be added.

In the following section, we present an analysis of the variability of maximum precipitation and temperature series on an annual basis to see the correlation between the projected hydrological extremes and the input climate extremes. In Fig. 2, raw annual maximum daily precipitation and temperature time series for the Biala Tarnowska catchment obtained from the seven GCM/RCM models under the RCP4.5 scenario are shown. The periods cover the whole length of historical and projected years (1971-2100). The upper panel of Fig. 2 presents annual minimum precipitation based on corrected precipitation projections (the upper panel), annual maximum precipitation based on raw projections (middle panel) and temperature mean projections for corrected data are presented in the lower panel of Fig. 2.

The results show a visible increase of the annual maximum temperature and an increase of temporal variability with time, in particular for the maximum precipitation values from 2016 onward.

The English writing style and grammar

The English and grammar was, and will be, checked by a native English speaker.

Specific comments

RC1. P1, L7-9: It is not clear what is meant with a 'multi-model approach' and which steps are followed. AC1. The 'multi-model approach' is an approach which considers multiple climate models and multiple hydrological parameter sets. To avoid possible confusion this wording will be changed: "The approach followed is based on . . ..”

RC2. P2, L9-11: The first question probably is related to the magnitude of the uncertainty, since this is still largely unknown and not systematically investigated. AC2. The sentence will be changed to: The question arises as to how large the uncertainty is and if it is acceptable to the end-user in adaptations to climate change and flood and drought risk assessments.

RC 3. P2, L15-16: "…..can never be accurately evaluated …." is a very strong statement, please rephrase. AC3. The sentence will be rephrased to: "However, complex hydrological and climate models are difficult to be accurately evaluated, because of uncertainty in observations, parameters and model structure simplifications."

RC 4. P2, L24-P3, L2: The authors mainly consider hydrological model and parameter uncertainty in their review. It might be worthwhile to firstly give an overview of all uncertainties involved in this type of studies including a classification. One such classification could be input, (hydrological) model system and output, and the literature can be reviewed accordingly. Now, uncertainties in the input (scenarios, GCMs, RCMs, downscaling, initial conditions etc.) are hardly reviewed. A complete overview of the uncertainties will also enable a better justification of the uncertainty sources considered in this study (see also page 3, lines 4-5). AC 4. As already discussed, the influence of other sources of uncertainty, including the choice of emission scenario, climate models (GCM/RCMs), downscaling and catchment type was performed by Osuch et al (2016) using a case study that included the catchment used in this paper. This paper was focused on predictive hydrological uncertainty to show that different objective functions should be applied when high and low flow extremes are considered. Apart from hydrological model parameters, seven climate models were also used and the spread relating to extreme index distribution was taken into account. However, the reviewer made the important point that our aims were not clearly enough presented and that the review of different sources of uncertainty would help to improve the presentation of that aim considerably. This part of the paper will be changed to justify better the choice of those three sources of uncertainty.

RC 5. P3, L14-15: The question is whether you can determine the uncertainty due to the choice of the extreme value distribution ('distribution fit') using time series of different lengths. When assessing effects of time series with different lengths on the results you might get an estimate of the influence of data quantity on the uncertainty in the results, but not of the influence of the goodness-of-fit of the distribution on the uncertainty. Furthermore, it seems only part of the statistical uncertainty is assessed

in this way, since for instance the influence of different extreme value distributions and extrapolation uncertainty is not taken into account. AC 5. Thanks to the reviewer, it is good point. However, we did not use different lengths of time series in order to determine the uncertainty due to the choice of the extreme value distribution. The sentence was misunderstood. In order to make our presentation more clear the sentence should read as follows: The uncertainty related to the distribution fit is analysed in two stages, using, separately, two different lengths of flow record to derive the quantiles of maximum and minimum annual flows, the 30-year long and 130-year long time series of future flow projections. The popular method of a comparison of changes in flow quantiles between the reference period and future periods is based on relatively short (e.g. 30-year) periods. It is well known that an extrapolation of a distribution function based on 30-year long time series towards 1-in-100 year quantiles involves very large errors (Strupczewski et al. 2011). Even the estimates of 1-in-30 year quantiles based on the 30-year long data are biased with large errors. We compare these errors with those involved on 1-in-100 year estimates obtained using the 130 year long time series. The question we pose is whether the estimates of future trends of extreme indices and their relative changes can be useful at all in view of the uncertainties involved.

RC 6. P3, L29-30: How many precipitation stations have been used to assess the catchment average precipitation (assuming lumped hydrological modelling has been carried out)? Has any elevation (or other) correction been incorporated? AC 7. We used five gauging stations to derive aerial precipitation in the catchment using Thiessen polygons. We did not use any elevation correction in this paper. However it was applied to the same catchment by Benninga et al (2017) and showed that the increase in precipitation due to the elevation is about 3%. RC 7. P4, L11: An important uncertainty source in climate impact studies is the uncertainty due to greenhouse gas emission scenarios. Hence, a limitation of this study is the use of only one emission scenario (RCP4.5) while one would expect the use of at least two scenarios (which are available in EURO-CORDEX). At least the authors should explain the implications of this limitation for their results. AC 7. The RCP 4.5 was applied because it is a stabilization scenario and thus assumes the imposition of emissions mitigation policies. The RCP 4.5 is derived from its own "reference", or "no-climate-policy", scenario. This reference scenario is unique to RCP 4.5 and differs from RCP 8.5, RCP 6.0 and RCP 2.6 (Smith and Wigley 2006; Clarke et al. 2007; Wise et al. 2009). The influence of the emission scenario on flood indices was studied by Osuch et al. (2016) whilst the low flows were analysed by Osuch et al (2017; submitted for publication). Both those studies indicated that emission scenario choice has a relatively small influence on the results. The implication of the choice of only one emission scenario will be explained in the revision.

RC 8. P5, L9: Why is QM applied in this study? The reasoning behind this choice is not completely clear from the preceding sentences. AC 8. Many popular existing bias correction methods have been reviewed and compared and quantile mapping (QM) was found to outperform other methods (Gudmundsson et al., 2012; Teutschbein and Seibert, 2013; Chen et al., 2013; Osuch et al., 2016). More recently, the standard non-parametric QM method has been adapted to more explicitly preserve the raw modelled climate change signals (Willems and Vrac, 2011; Sunyer et al., 2014; Cannon et al., 2015). This means, in the QM method, that a raw modelled value is always corrected by the same value of bias or error that is determined by its respective quantile in the reference period.

RC 9. P5, L18-19: Did Osuch et al. (2015) model the same catchment as in this study and therefore, can it be assumed that the same five parameters are sensitive? And are the same five parameters sensitive for low flows and for high flows? That would be remarkable. AC9. The HBV model was applied in different hydro-climatic condition in Poland by different researchers and they found the five most sensitive parameters for both high flow and low flow characteristics. The set of five parameters chosen in this study was dictated by the most common catchment conditions. Therefore it is not surprising that the same parameters are sensitive in both high and low flow conditions. However, in this study we used two objective functions to encapsulate the high and low

flow characteristics instead of selected best parameters only belonging to low flow and high flow.

RC10. P6, L15-16: How many Monte Carlo simulations have been executed and is this number sufficient (compare with literature)? AC 10. 20000 MC simulations were executed. Many research papers recommend above 10, 000 MC (e.g. Xiaoli Jin et al., 2010; Romanowicz et al., 2013; Houska T. et al., 2014).

RC 11. P6, L22: Is it common practice to determine the thresholds in an iterative way? The determination of the threshold based on the requirement that 95% of the observations should be in the 95% confidence interval seems to be reasonable. However, please refer to other studies employing the same approach. AC 11. To our knowledge, it is a common practice. The thresholds determine the variance of the predictions. Too high a threshold results in too narrow confidence bands. By iteration we meant the "trial and error approach" which does not involve any algorithm. We would be surprised if the iterative determination of threshold values has not yet been introduced, but we are not aware of any studies that have followed this approach. We will change the wording to avoid confusion.

RC 12. P7, L4-5: In general it is doubtful whether distributions with a 'large' number of parameters will model data in a more accurate way than distributions with a small number. This partly depends on the data quantity and quality and similarly as in hydrological modelling there will be a balance between the complexity of the distribution (i.e. number of parameters) and the amount of data (and quality). AC 12. We agree with the reviewer that there must be a balance achieved between the complexity of the distribution (i.e. the number of parameters) and the quality of data. We admit that this sentence can be deleted as it is a too large generalization.

RC 13. P7, L7-8: What does an 'overall good performance' mean? Compared to which other distributions? AC 13. A number of distributions was tested including a three-parameter lognormal and an inverse Gaussian. GEV was the only distribution

that performed well both for the high and low flow extremes. Although it was not necessary to use the same distribution for both extremes, it made our discussion more transparent.

RC 14. P7, L25-27: It is not completely clear why the analyses are performed for a period of 130 years. Since the manuscript is about impacts of climate change on hydrological extremes, you would expect a comparison between historic and future climate conditions. Furthermore, climate change automatically implies the existence of nonstationarity and as such, by considering a period of 130 years assuming stationarity by using the same extreme value distribution will result in serious flaws. AC 14. We do not think that the impact of climate change on hydrological extremes should be based on a comparison between historic and future climate conditions. What we propose here is to study the trend of projected indices instead of the "change". The Biala Tarnowska catchment does not show any non-stationarity in the extreme flow events (Meresa et al. 2017, submitted for publication). Therefore it is a suitable catchment to compare both approaches. We are aware that non-stationary flood frequency analysis has to be applied for non-stationary extreme events. We want to show here that taking 30-year long time-series to compare between reference and future periods involves large uncertainty even for 30-year return flows. The uncertainty ranges of 30-year return period flows obtained using 130-year long time series can be nearly four times smaller.

RC 15. P8, L7-12: The idea behind this section is not clear. Why is the trend in daily annual maximum precipitation and temperature analysed while the interest is in uncertainty in hydrological indices with climate change? Moreover, why is the daily annual maximum precipitation of interest and not for instance the two-day or three day precipitation (which might be stronger correlated to annual maximum discharge values)? Which temporal resolutions of precipitation are relevant for annual minimum flows? And what is the supposed role of daily annual maximum AC 15. The idea behind presenting the precipitation and temperature patterns was to show the variability of driving forces behind the changes in flow extreme index. However, the idea was not properly

explained and followed. For a catchment of that size, daily maximum and mean sums of precipitation are well correlated with the flow patterns. The temperature patterns, on the other hand, present the changes in the evaporation losses and possibly, indicate the changes in flood regime.

RC 16. P8, L14-20: How have the different criteria for high and low flows been applied in continuous hydrological modelling for periods of 30 years (or more)? When is the 'high flow' parameter set being used and when the 'low flow' one? What is the threshold for low flows and high flows; a specific discharge value or exceedance frequency? AC 16. As explained in section 3.3, we applied a stochastic formulation to the estimation of the HBV model parameters. That means that 20000 simulations of the HBV model were run for the 30-year long calibration period with parameters sampled randomly within the assumed parameter ranges. The calibration is performed using the observed precipitation, temperature and flow records. We applied logNSE criterion for low flow and NSE criterion for high flow index to all the simulated flow series. Then we evaluated thresholds for the criteria, called likelihood thresholds, based on the requirement of 95% of the observations should be in the 95% confidence interval separately for high and low flows (Table 3). In other words, we do not have one "high" or "low" parameter set but we have two multiple sets (each including thousands of parameter sets) representing "high" and "low" flow models.

RC 17. P9, L5: Which best parameter sets are meant here? When is the best low flow parameter set used and when the best high flow parameter set? AC 17. Results shown in Fig. 4 were obtained from the HBV model simulations fed by the precipitation and temperature projections obtained from the seven GCM/RCM models under the RCP4.5 scenario for the best parameter sets from the MC parameter samples, giving the highest weights derived from the NSE for the high flows and logNSE for the low flows, respectively. The raw hydro-meteorological projections were applied to study the high flow index whilst bias corrected precipitation and temperature data were used for the low-flow index studies.

RC 18. P9, L7-8: 'twice as large'; where do we see that? AC 18. Sentence will be corrected to: Obtained flow projections shown in Fig. 4, follow the rainfall projections shown in Fig. 2, with annual maximum flow values even four times larger than historical events occurring after 2016 for some GCM/RCM model projections.

RC 19. P9, L12-22: This evaluation is not clear to me. Why do the authors evaluate results at a monthly scale? How can you assess annual maximum flows for each month? What do the authors mean with 'range' of annual maximum flows? AC 19. Thank you, it is corrected in the main manuscript, "annual" was replaced by "monthly". Analysis of variability of monthly flows was presented to illustrate seasonal changes of extreme flows in the near future period and the uncertainty related exclusively to hydrological model uncertainty for each climate model projection. Some changes of seasonality are visible for high flows, but low flows do not show any distinctive differences between reference period and near future. We agree with the reviewer that changes in seasonality are not explored further and therefore this section can be removed, together with Fig. 5.

RC 20. P10, L9-10: The decrease in the spread of Q30 in the far future compared to the near future is strange. The authors should reflect on this. Is it related to the fact that only one RCP scenario is taken into account? AC 20. This smaller spread of the far-future projected changes was also observed in the other climate impact studies on the same catchment (Osuch et al., 2017) for both the RCP4.5 and RCP8.5 emission scenarios using the HBV model for hydrological simulations. Research is on-going to explain that phenomenon.

RC 21. P10, L20-22: Also this observation needs discussion. Why the spread is more evenly distributed for minimum flows compared to maximum flows? AC 21. This is related to the influence of the climate model spread on the simulations (Osuch et al., 2017). It is much bigger for high flows and not very big for the low flows. We also have to remember that low flow simulations used bias-corrected meteorological drivers whilst the high-flow simulations were driven by the raw data. Bias correction decreases

the variability of climate models.

RC 22. P11, L13-14: Are the relative differences for annual minimum flows also smaller? AC 22. Yes. The sentence should read: The relative differences obtained for the annual minimum flow Q30 estimates are smaller, suggesting that low flow quantiles are less susceptible to the errors related to the length of the evaluation period.

RC 23. P12, L7-9: This is an interesting topic, but has not been investigated in this study since only one catchment has been considered. AC 24. We agree with the reviewer, this sentence is out of context and should be deleted.

RC 24. P12, L11-14: This is an interesting result assuming that all methodological steps are logical and correctly carried out. What is the reason for the importance of uncertainty due to climate models for high flow and the important of hydrological model parameter uncertainty for low flows? This is very important and interesting to discuss. AC 24. The important role of hydrological model uncertainty in low flow predictions was already noticed in forecasting (Beninga et al., 2017). That effect can be explained by the ratio of the prediction noise (in this case described by the hydrological model uncertainty) to the input signal which is much higher for low flows.

RC 25. P12, L23-24: What do the authors mean with 'this allows the problem of nonstationarity of model parameters to be avoided'? AC 25. The sentence should read: (iii) conditioning of the hydrological model was performed using different criteria for low and high flows in order to ensure the best model fit for the extremes; this does not solve the problem of non-stationarity of model parameters but allows for focusing on parameter sets adequate for low or high flow regimes.

RC 26. P12, L29-31: This statement seems to be obvious; the larger the ratio of return period vs. data length the higher the uncertainty. However, this extrapolation uncertainty is not explicitly assessed in this manuscript. AC 26. Thank you for the comment. This statement should read "analysis of the influence of length of time series records on the uncertainty bands of the high flow quantile estimates and their changes

suggests that the ranges of quantiles of return periods Q30 are up to four times smaller when the long-term flow projections are used (Table 4). The low flow Q30 quantiles are less influenced by the length of the record.

RC 27. P23, Table 2: The ranges defined by the lower and upper bounds frequently do not match with the optimal values (e.g. for ALFA, PERC, CLFUX). Can you explain this? Furthermore, some lower and upper bounds are exactly the same. Does this indicate that these parameters are deterministic? What about CFMAX (not mentioned as sensitive in section 3.3)? Finally, an upper bound of 2 for LP is impossible and an optimum value of 1 is remarkable at least (it would mean only potential evapotranspiration under fully saturated conditions). AC 27. We thank the reviewer for this comment. There was a mistake in Table 2. The HBV model was calibrated using GLUE and optimal parameter sets were derived in the form of multiple parameter sets, different for the high and low flows. Therefore this table should not include the "optimal" parameter values. The Table 2 will be corrected

Technical corrections

RC 1. P1, L11: What is the distribution fit? AC 1. 'distribution fit' will be changed into "theoretical distribution fit error"

RC 2. P1, L13: What kind of weighting do the authors mean? AC 2. "with a separate criterion for high and low flow extremes"

RC 3. P1, L16: What is the difference between climate model variability and climate projection ensemble spread? Please use a consistent terminology. AC 3. The meaning of "variability" is not the same as "spread". Here we meant "variability".

RC 4. P2, L3: What is inverse modelling in this respect? Is this term commonly used for calibration and validation purposes based on observed (historic) data? AC 4. "Conditioning" can be used here instead of "inverse modelling", if it is clearer. Inverse modelling refers to model parameter calibration based on historical data.

RC 5. P2, L6: "weighting" instead of "weighing". AC5. Corrected: "weighting" instead of "weighing".

RC 6. P3, L8: What is the 'relevant variability' of extreme index estimates? AC 6. Changed into "a direct assessment of variability of extreme index estimates"

RC 7. P3, L19: The case study has already been mentioned. AC 7. Thank you, the sentence will be deleted.

RC 8. P3, L30: The maximum daily precipitation? During which period? AC 8. Thank you, corrected. 'Maximum precipitation was 68.3 mm d-1 and annual mean streamflow is 0.4 m3s-1 over the observation period.' Changed to 'The annual maximum precipitation, annual minimum streamflow and annual mean streamflow of the catchment were 68.3 mm, 0.4 m3s-1 and 5.43 m3s-1 respectively over the observation period (1971-2000)'.

RC 9. P3, L30-31: Which period for the streamflow) Isn't 0.4 m3/s a very low value for catchment area of about 1000 km2? AC 9. Thank you. It is the same as with the previous comment. It is a minimum streamflow.

RC 10. P4, L12-14: Why do the authors use these complex abbreviations for the GCM/RCM combinations? It is not clear what the meaning of all the numbers is. Try to be consistent with the descriptions in Table 1. AC 10. The Table will be corrected

RC 11. P5, L12: Do you have a reference for the Matlab version of HBV? AC 11. The MATLAB version of HBV used in this study was based on Lindstrom et al (1997). The original MATLAB code from Twente University NL, was further developed and adjusted for the purpose of climate impact studies in the Institute of Geophysics PAS.

RC 12. P5, L15-17: Only 12 out of 14 HBV parameters are mentioned. In which routines can we find CFLUX and PERC (see line 19)? AC 12. Thank you. It is changed to 'These routines are governed mainly by fourteen HBV parameters, of which, six (TT, TTI, CFMAX, DTTM, CFR, WHC), three (FC, LP, BETA, CFLUX), two (KF, ALPHA) and

one (KS, PERC) parameters are representing each routine respectively.'

RC 13. P5, L17: 'routines' instead of 'routing stage'? AC 13. Thank you; corrected: 'routines' instead of 'routing stage'?

RC 14. P6, L24-P7, L3: This general description of the GEV distribution is not necessary here and can be found in many text books. AC 14. It will be deleted

RC 15. P7, L16-17: What do the authors mean with ": : : aggregated speared of flow quantile change : : :"? AC 15. By "aggregated speared of flow quantile change" we meant "integrated spread . . . ".

RC 16. P7, L19: 'squared' instead of 'squere'. AC 16. Thank you, it is corrected in the main manuscript.

RC 17. P7, L22: The title suggests that the results of this study will be described. Please rephrase the title. AC 17. "Description of the results" would be better?

RC 18. P7, L23: Different temporal resolutions? Shouldn't it be different lengths of data periods? AC 18. Agree: 'different temporal resolutions' will be changed to 'different lengths of data periods'

RC 19. P7, L18: The meaning of all variables should be explained in the text. AC 19. All the variables will be explained. Where: $T\_SS_{ijk}$ is total sum square error for the specific hydrological extreme indicator (e.g. relative change in Q30) for the ith parameter sets range, jth climate model and kth distribution parameter range and $\mu$ is the overall mean and $\varepsilon\_ijk$ denotes the white Gaussian error.

RC 20. P8, L6: "Results and discussion"? AC 20 Thank you, it is corrected in the main manuscript. As "Discussion of the results"

RC 21. P9, L2: 'the 10-year moving average from the ensemble mean'? AC 21. Thank you, it is corrected in the main manuscript. Corrected as 'the 10-year moving average from the ensemble mean' changed to 'mean from the ensemble of seven climate

models'

RC 22. P9, L15-16: Fig. 5a is mentioned twice. AC 22. Fig. 5 will be deleted

RC 23. P9, L29-30: Decreases in minimum flows and increases in maximum flows? Shouldn't it be the other way around (according to the caption of Fig. 6)? AC 23. Thank you, it is corrected in the main manuscript: 'Figure 6. Empirical flow quantiles of annual maximum flow (upper panels) and annual minimum flow (lower panels) under baseline and future climates (near and far future periods); the climate model spread is presented as a shaded area; green line denotes the mean value from all the GCM/RCM model realizations, red line denotes the averaged results obtained for the reference period.'

RC 24. P10, L6-7: Here, the annual minimum flows increase (see previous comment). AC 24. Thank you. It is corrected as previous comment

RC 25. P10, L9: What is Q30? Commonly, that is a discharge with a non-exceedance frequency of 30%. However, here it seems to be an annual maximum flow with a return period of 30 years? AC 25. Yes, it is annual maximum flow with a return period of 30 years. For annual maximum flow those two definitions have the same meaning. However, now to avoid confusions, we used as Qt30 in the main manuscript.

RC 26. P11, L7: 'Table 4' instead of 'Table 3'. AC 26. Thank you, it is corrected in the main manuscript. 'Table 4' instead of 'Table 3'.

RC 27. P11, L26-P12, L2: The first part of the conclusions can be omitted (can be part of introduction section). AC 27. Thank you, it is corrected in the main manuscript. Deleted

28. P12, L9: 'hydrological parameter uncertainties' instead of 'hydrological model uncertainties'? AC 28. Thank you, it is corrected in the main manuscript. Corrected to 'hydrological parameter uncertainties' instead of 'hydrological model uncertainties'

RC 29. P12, L24-27: This is a repetition of lines 11-14. AC 29. Thank you, it is corrected in the main manuscript. Deleted

RC 30. P13, L3: A paper in preparation should not be included in the reference list. AC 30. This paper has already been submitted: "Meresa, H., Romanowicz, R.J., Napiorkowski, J.J.: Trends of projections of hydrological extreme indices in the 21st century, submitted, 2017." RC 31. P13-17: The reference list and referencing contain many errors, typos and inconsistencies. This should be carefully and thoroughly double-checked. AC 31. The reference list will be corrected.

RC 32. P18, Fig. 1: What is the unit of the DEM map? AC 32. The unit is meter. Thank you, it is corrected in the main manuscript.

RC 33. P18, Fig. 2: The interquantile range of what? Of the seven GCM-RCM combinations? In that case it would be better to show the individual model results, i.e. one annual maximum for each combination so 7 points per year. AC 33. Figure 2. Annual maximum daily precipitation and mean air temperature time series for the BialaTarnowska catchment obtained from the seven GCM/RCM models under the RCP4.5 scenario; upper panel: projected raw annual maximum daily precipitation; lower panel: projected corrected annual mean daily temperature in the 1971-2100 period; boxplots show interquartile range of 75th and 25th; whiskers show 50th percentiles. We think that boxplots are better suited to present the precipitation and temperature variability than individual model results.

RC 34. P19, Fig. 3: In particular the scale of the upper panel looks strange. Flows in cubic mm? How accurate is your model? Please use the same (realistic) x-axis ranges. AC 34. We guess that the reviewer means Fig. 4. The y-axis units should be in cubic meters per second. The figure will be corrected.

RC 35. P19, Fig. 4: This figure (and also Fig. 2) is too small. What do we see here? AC 35. Fig. 4 presents projected annual maximum and minimum flow. The figures 2 and 4 will be enlarged.

RC 36. P20: The differences between historic and future periods cannot be clearly seen in these figures. AC 36. Following the first reviewer comments we decided to

remove Fig.5 together with the subsection 4.4.

RC 37. P21, Fig. 6: What are the different lines in these figures? And is baseline and reference period the same? AC 37. Thank you, it is corrected in the main manuscript. Changed to 'Fig. 6. Empirical flow quantiles of annual maximum flow (upper panels) and annual minimum flow (lower panels) for the reference period and future climates (near and far future); the climate model spread is presented as a shaded area; green line denotes the mean value from all the GCM/RCM model realizations in each period (near and far future period), red line denotes the averaged results obtained for the reference period. Each black line represents individual climate models'

RC 38. P21, Fig. 7: In the caption 'right hand panel' is mentioned twice. AC 38. The figure caption will be changed to: Fig. 7. Total uncertainty ranges of theoretical GEV-based annual maximum (left hand panels) and minimum (right hand panels) flow quantiles over 30 year periods; upper panels - the reference period 1971-2000; middle panels - near future 2021-2050; lower panels - far future 2071-2100. RC 39. P22, Fig. 8: Idem, annual minimum flow is mentioned twice. AC 39. Figure 8 captions changed to: Fig. 8. Total uncertainty ranges of flow quantiles over 130 years period; annual minimum flow as a function of return level (right hand panel) and annual maximum flow as a function of return level (left hand panel), based on a GEV distribution fitted to the projected annual flow (1971-2100).

RC 40. P23, Table 1: Which meteorological institute is connected to RACMO? AC 40. Netherlands.

RC 41. P23, Table 2: The caption is not clear. AC 41. Changed to 'Table 2. HBV parameter ranges: upper band (UB), lower band (LB) value' RC 42. P24, Table 4: What do the authors mean with 'change in width of . . .'? What compared to what? AC 42. Table 4. Change in the width of 0.95 confidence intervals for Q30 for annual maximum and minimum flows evaluated using different time periods of different lengths, first three columns corresponding to Fig. 7 (30-year period) and the fourth column corresponding

to Fig. 8 (130-year period).

References Benninga, J-H., Booij, M., Romanowicz, R.J., Rientjes, T.H.M., 2016, Performance of ensemble streamflow forecasts under varied hydrometeorological conditions, Hydrol. Earth Syst. Sci. Discuss., doi:10.5194/hess-2016-584, 2016. Blasone, R., Vrugt, J., Madsen, H., Rosbjerg, D., Robinson, B., & Zyvoloski, G. (2008). Generalized likelihood uncertainty estimation (GLUE) using adaptive Markov Chain Monte Carlo Sampling. Advances in Water Resources(31), 630-648. Meresa, H., Romanowicz, R.J., Napiorkowski, J.J.: 2017, Trends of projections of hydrological extreme indices in the 21st century, Acta Geophysica, submitted, for publication Osuch, M., R.J. Romanowicz, W. Wong, 2017, Analysis of low flow indices under varying climatic conditions in Poland, Hydrology Research, submitted for publication

---

## Author Comment (AC2) · 9 Mar 2017

Dear Editor and Reviewers,

Thank you very much for the constructive comments that will help to considerably improve and clarify the manuscript. The reviewers, put enormous effort into proof-reading our paper line by line and trying to clarify all the less-than-satisfactory statements and mistakes. We believe that the input will improve the manuscript significantly. All comments have been addressed point-by-point. Following the reviewers' feedback we will make the corresponding changes in the manuscript. Reviewers' comments are in italics.

Anonymous Referee #2

Overview

RC. The authors assess the effect of different uncertainty sources on climate change projections. The presentation of the results is easy to follow and interpret. Especially Figure 9 is very informative. However, there is room for improvement using specific comments and checklist below. I recommend major revision as the model calibration part is not clear. AC. We thank the reviewer for concise and valuable comments.

Specific Comments:

RC1. Table 2: Optimal values of some parameters are out of lower and upper limits e.g. CFMAX which cannot be reached by an algorithm e.g. SCEUA, CMAES etc. How was this achieved by a calibration algorithm? Did you follow a manual calibration scheme? AC1.The table 2 is now corrected (the new version is given in AC. 27 of the answers to the 1st reviewer). We do not use deterministic calibration, instead the GLUE -based stochastic calibration is applied. That table was put by mistake.

RC2. Demirel et al (2013a) is in the reference list but not in the text. AC2. Thank you, it has been corrected

RC3. Please explain the abbreviations used at legend in figure caption. The legend of Fig8 is confusing: "distn"? AC3. Thank you, it has been corrected. "distn" replaced by "distribution"

RC4. Did you compare uncertainty in HBV model parameters with other studies (Addor et al., 2014; Demirel et al., 2013b; Osuch et al., 2015) using HBV model for forecasting hydrological extremes? How would the results overlap for 10 day forecast (Demirel et al., 2013b) and long term climate predictions in EUROCORDEX (dataset used in this study)? AC4. The uncertainty in HBV model parameters were compared with the other studies, including Osuch (2015) and Demirel et al (2016b). The latter explored the influence of uncertainty in input, hydrological model parameters and initial conditions

on 10-day ensemble flow forecasts. The results show that parameter uncertainty has the largest effect on the medium range low flow forecasts, which is consistent with the present paper findings. Addor et al. (2014) concentrated on the influence of different hydrological model structure, involving three hydrological models, emission scenarios, climate models, post-processing and catchments. Their results indicate that influence of model structure varies with the catchment. However the authors did not take into account hydrological model parameter uncertainty, which is the main focus of the present paper. Osuch (2015) compared three sensitivity analysis techniques to describe the HBV model parameter interactions. We used the output of that paper to eliminated less sensitive HBV model parameters in order to minimize computational cost.

RC5. Fig5: Parameter uncertainty should be presented differently to assess the contribution of each parameter uncertainty to total uncertainty. From this figure the reader can't see the most uncertain parameter. A figure similar to Figure 4 in Demirel et al (2013b) or Fig9 in the current manuscript can be very useful for modelers. This can be easily done as the GLUE results would allow such ranking. AC5. Thank you for the comment. We decided to remove this figure and subsection 4.4 following the first reviewer comments.

RC6. Conclusion 2 (ii): Please explain the drizzle effect? Not clear. AC6. Simulated climate variables (precipitation and temperature) by individual GCMs/RCMs often do not reach agreement with observed climate time series. This is due to the effect of systematic and random model errors of GCMs/RCM simulations. Such systematic errors lead to simulate many drizzle days (i.e., too many days with very low precipitation intensity and too few dry days). The drizzle effect is related to the performance of climate models. It presents itself in the form of frequent rainfall of a very small intensity. The physics behind precipitation generation is very complex and involves processes operating on a wide range of scales. The frequent 'drizzle' is produced mainly by convective parameterization. It appears in many climate models and invokes errors in the intensity and frequency of precipitation (Terai et al. 2016). The correction can be

performed using the number of wet days in a month (Osuch et al. 2016). Because of this bias in precipitation, using direct climate model output as inputs to hydrological modelling for low flow analysis often leads to unrealistic results and therefore bias-correction is required in the case of low flow projections.

RC7. Section 3.6 and Conclusion 5 (v): Is ANOVA method a global or local sensitivity analysis method? Can interactions (parameter etc) be assessed using this method? Why ANOVA is used instead of other elementary and global methods e.g. Morris, SOBOL, PEST, FAST etc. These aspects of the ANOVA method should be described in section 3.6 and conclusions should follow these details. AC7. Nowadays, many global sensitivity methods have been proposed and used, such as Fourier amplitude sensitivity test (FAST), Regional Sensitivity Analysis (RSA), Analysis of Variance (ANOVA), Parameter Estimation Software (PEST), Morris, and Sobol method. Among these global sensitivity analysis methods, ANOVA is proved to be one of the most robust and effective tools to analyze both continuous and discrete factors (Montgomery, 1997), and it is widely applied in hydrology (Bosshard et al., 2013; Zhan, et al., 2013; Lagerwalla, et al., 2014; Addal et al.,2014; Giuntoli et al., 2015; Osuch, 2015). We used ANOVA approach due to its numerical facility (MATLAB) and ability to evaluate the main and interactive effects between factors considered.

RC8. Conclusion bullets are confusing. Two times "iv" exists and sentences are not clear. There are typos too. For example Conclusion vi should start with capital. Please rephrase them with short and clear conclusions. And relate them to the results section. Bullet conclusions in Demirel et al (2013b) can be an example. For each result section one paragraph is given in conclusion. AC8. Thank you, it is corrected in the main manuscript.

References

- Bosshard, T. Carambia, M. Goergen, K. Kotlarski, S. Krahe, P. Zappa, M. and Schar, C. Quantifying uncertainty sources in an ensemble of hydrological climateimpact projections. WATER RESOURCES RESEARCH, VOL. 49, 1523-1536, doi:10.1029/2011WR011533, 2013 - Montgomery, D.C. (1997). Design and Analysis of Experiments. Wiley and Sons Ltd.: New York. - Zhan, Y.; Zhang, M. Application of a combined sensitivity analysis approach on a pesticide environmental risk indicator. Environ. Model. Softw. 2013, 49, 129-140. - Lagerwalla, G.; Kiker, G.; Muñoz-Carpena, R.; Wang, N. Global uncertainty and sensitivity analysis of a spatially distributed ecological model. Ecol. Model. 2014, 275, 22-30. - Addor, N., O. Rossler, N. Koplin, M. Huss, R. Weingartner, and J. Seibert (2014), Robust changes and sources of uncertainty in the projected hydrological regimes of Swiss catchments, Water Resour. Res., 50, 7541-7562, doi:10.1002/2014WR015549. - Giuntoli, J.-P. Vidal, C. Prudhomme, and D. M. Hannah. Future hydrological extremes: the uncertainty from multiple global climate and global hydrological models Earth Syst. Dynam., 6, 267-285, 2015. doi:10.5194/esd-6-267-2015 - Osuch M. Sensitivity and uncertainty analysis of precipitation-runoff models for the middle Vistula basin, in GeoPlanet: Earth and Planetary Sciences, 2015. DOI:10.1007/978-3-319-18854-6-5.
* * *

---

## Author Comment (AC3) · 13 Mar 2017

Hadush K. Meresa and Renata J. Romanowicz

hadush@igf.edu.pl

Dear Editor and Reviewer,

This is the corrected version of the response to the reviewer comments. Please ignore the previous version which was not properly formatted (the lines merged). We also added the tables 1 and 2.

Thank you very much for the constructive comments that helped to considerably improve and clarify the manuscript. The reviewer, put enormous effort into proof-reading our paper line by line and trying to clarify all the less-than-satisfactory statements and mistakes. We believe that the input will improve the manuscript significantly. All comments have been addressed point-by-point. Following the reviewers' feedback we will make the corresponding changes in the manuscript.

Anonymous Referee #1

RC1. General comments

This paper is about the uncertainty of extreme flows with climate change. For that purpose, the authors use seven combinations of global climate models (GCMs) and regional climate models (RCMs) with one greenhouse gas concentration scenario to represent uncertainty in climate change. Furthermore, they use the GLUE method to represent hydrological parameter uncertainty and uncertainty in extreme value distribution parameters to represent the uncertainty in the statistical extreme value distribution. These three sources of uncertainty are investigated using the HBV hydrological model applied to a medium-sized Polish catchment. Although the topic is interesting and relevant for this journal, the paper is moderately written, lacks clarity in parts of the methodology and only briefly discusses results and insufficiently puts outcomes into perspective. For instance, the seemingly arbitrary choice to consider the three uncertainty sources is not justified. Are these three sources the most important ones or the easiest ones to quantify? Furthermore, the uncertainty due to the use of a particular extreme value distribution is not clearly and completely incorporated. A final example is the presentation and analysis of results, such as the analysis of annual maximum precipitation and temperature in relation with annual maximum flows and in particular annual minimum flows. In this case and several other cases it is not always clear which results are shown, why they are shown and what can be concluded from the results. Many other specific (and important) comments can be found below. Furthermore, the English writing style and grammar is moderate (including several typos); some examples can be found in the section 'technical corrections'.

AC1. General answer

Following the reviewer's general and specific comments, the clarity of the methodology

will be improved and the outcomes will be described in a wider perspective.

The choice of three particular sources of uncertainty, namely, a set of climate model ensembles, hydrological model parameter uncertainty and uncertainty in fitting extreme value distribution, was dictated by one of the aims of the research – i.e. an assessment of influence of hydrological model uncertainty on projections of low- and high-flow extremes and the relative contribution of that "predictive" uncertainty in the spread of extreme indices related to the climatic model spread and the distribution fitting error.

This choice followed from a discussion on all the sources of uncertainties and a review of research done so far on the assessment of uncertainty of projections of hydrological extremes. The "predictive" model uncertainty is the only one which can be decreased when conditioned on the observations. The other sources of uncertainty have an "epistemic" nature and cannot be decreased. Bearing in mind the aims of the study, we restricted the sources of epistemic uncertainty to those which have the largest impact – i.e. climate model spread, omitting the uncertainty related to bias correction or geography. The error related to the distribution fit was included as an essential part of the extreme index evaluation, which requires extrapolation of annual maximum or minimum flow distributions to higher order quantiles (e.g. 1-in-100 year, or 1-in-200 year). In this paper, the error related to the evaluation of maximum and minimum annual flow statistic was treated as epistemic, that means, not conditioned on real observations.

In addition, Osuch et al. (2016) presented the influence of emission scenario, climate model, bias correction method and geography on flow indices in a case study that included the same catchment, Biala Tarnowska. Therefore we wanted to avoid the repetition. In this regard, our paper is an extension of the former paper, focusing on the influence of hydrologic model uncertainty on annual maximum and minimum flow projections. In our opinion, including the other sources of uncertainty would obscure our aim.

The choice of extreme value distribution followed the validation of suitability of this

distribution to describe the projected annual maximum and minimum flows using the probability plots. The MATLAB- based GEV distribution fitting algorithm was applied to all the climate models and the a posteriori hydrological model parameter set. This algorithm provides the estimates of 0.95 confidence bands for the distribution parameters. These parameters were subsequently used to obtain upper and lower confidence bands of the distribution through the inverse GEV model. In order to simplify the procedure, instead of sampling from the GEV parameters within the parameter space common to all hydrologic and climate model simulations, we sampled from each set of parameters assuming a normal distribution with the variance specified by the parameter upper and lower 0.95 confidence value, and in addition, assuming the independence of the GEV model parameters. The obtained 0.95 GEV distribution confidence values were used to estimate the spread of results related to the distribution fit.

Bearing in mind the large number of simulations, it was not possible to choose the best distribution for each projected time series. Furthermore, the aim of this study is to assess the ranges of uncertainty of extreme indices rather than their exact values.

RC2. A final example is the presentation and analysis of results, such as the analysis of annual maximum precipitation and temperature in relation with annual maximum flows and in particular annual minimum flows. In this case and several other cases it is not always clear which results are shown, why they are shown and what can be concluded from the results.

AC2. We agree with the reviewer that the presentation should be much improved and clarified. The following explanation will be added.

In the following section, we present an analysis of the variability of maximum precipitation and temperature series on an annual basis to see the correlation between the projected hydrological extremes and the input climate extremes. In Fig. 2, raw annual maximum daily precipitation and temperature time series for the Biala Tarnowska catchment obtained from the seven GCM/RCM models under the RCP4.5 scenario

are shown. The periods cover the whole length of historical and projected years (1971-2100).

The upper panel of Fig. 2 presents annual minimum precipitation based on corrected precipitation projections (the upper panel), annual maximum precipitation based on raw projections (middle panel) and temperature mean projections for corrected data are presented in the lower panel of Fig. 2.

The results show a visible increase of the annual maximum temperature and an increase of temporal variability with time, in particular for the maximum precipitation values from 2016 onward.

The English and grammar was, and will be, checked by a native English speaker.

Specific comments

RC1. P1, L7-9: It is not clear what is meant with a 'multi-model approach' and which steps are followed.

AC1. The 'multi-model approach' is an approach which considers multiple climate models and multiple hydrological parameter sets. To avoid possible confusion this wording will be changed: "The approach followed is based on . . .."

RC2. P2, L9-11: The first question probably is related to the magnitude of the uncertainty, since this is still largely unknown and not systematically investigated.

AC2. The sentence will be changed to: The question arises as to how large the uncertainty is and if it is acceptable to the end-user in adaptations to climate change and flood and drought risk assessments.

RC 3. P2, L15-16: ". . ..can never be accurately evaluated . . .." is a very strong statement, please rephrase.

AC3. The sentence will be rephrased to: "However, complex hydrological and climate models are difficult to be accurately evaluated, because of uncertainty in observations,

parameters and model structure simplifications."

RC 4. P2, L24-P3, L2: The authors mainly consider hydrological model and parameter uncertainty in their review. It might be worthwhile to firstly give an overview of all uncertainties involved in this type of studies including a classification. One such classification could be input, (hydrological) model system and output, and the literature can be reviewed accordingly. Now, uncertainties in the input (scenarios, GCMs, RCMs, downscaling, initial conditions etc.) are hardly reviewed. A complete overview of the uncertainties will also enable a better justification of the uncertainty sources considered in this study (see also page 3, lines 4-5).

AC 4. As already discussed, the influence of other sources of uncertainty, including the choice of emission scenario, climate models (GCM/RCMs), downscaling and catchment type was performed by Osuch et al (2016) using a case study that included the catchment used in this paper. This paper was focused on predictive hydrological uncertainty to show that different objective functions should be applied when high and low flow extremes are considered. Apart from hydrological model parameters, seven climate models were also used and the spread relating to extreme index distribution was taken into account. However, the reviewer made the important point that our aims were not clearly enough presented and that the review of different sources of uncertainty would help to improve the presentation of that aim considerably. This part of the paper will be changed to justify better the choice of those three sources of uncertainty.

RC 5. P3, L14-15: The question is whether you can determine the uncertainty due to the choice of the extreme value distribution ('distribution fit') using time series of different lengths. When assessing effects of time series with different lengths on the results you might get an estimate of the influence of data quantity on the uncertainty in the results, but not of the influence of the goodness-of-fit of the distribution on the uncertainty. Furthermore, it seems only part of the statistical uncertainty is assessed in this way, since for instance the influence of different extreme value distributions and extrapolation uncertainty is not taken into account.

AC 5. Thanks to the reviewer, it is good point. However, we did not use different lengths of time series in order to determine the uncertainty due to the choice of the extreme value distribution. The sentence was misunderstood. In order to make our presentation more clear the sentence should read as follows: The uncertainty related to the distribution fit is analysed in two stages, using, separately, two different lengths of flow record to derive the quantiles of maximum and minimum annual flows, the 30-year long and 130-year long time series of future flow projections. The popular method of a comparison of changes in flow quantiles between the reference period and future periods is based on relatively short (e.g. 30-year) periods. It is well known that an extrapolation of a distribution function based on 30-year long time series towards 1-in-100 year quantiles involves very large errors (Strupczewski et al. 2011). Even the estimates of 1-in-30 year quantiles based on the 30-year long data are biased with large errors. We compare these errors with those involved on 1-in-100 year estimates obtained using the 130 year long time series. The question we pose is whether the estimates of future trends of extreme indices and their relative changes can be useful at all in view of the uncertainties involved.

RC 6. P3, L29-30: How many precipitation stations have been used to assess the catchment average precipitation (assuming lumped hydrological modelling has been carried out)? Has any elevation (or other) correction been incorporated?

AC 7. We used five gauging stations to derive aerial precipitation in the catchment using Thiessen polygons. We did not use any elevation correction in this paper. However it was applied to the same catchment by Benninga et al (2016) and showed that the increase in precipitation due to the elevation is about 3%.

RC 7. P4, L11: An important uncertainty source in climate impact studies is the uncertainty due to greenhouse gas emission scenarios. Hence, a limitation of this study is the use of only one emission scenario (RCP4.5) while one would expect the use of at least two scenarios (which are available in EURO-CORDEX). At least the authors should explain the implications of this limitation for their results.

AC 7. The RCP 4.5 was applied because it is a stabilization scenario and thus assumes the imposition of emissions mitigation policies. The RCP 4.5 is derived from its own "reference", or "no-climate-policy", scenario. This reference scenario is unique to RCP 4.5 and differs from RCP 8.5, RCP 6.0 and RCP 2.6 (Smith and Wigley 2006; Clarke et al. 2007; Wise et al. 2009). The influence of the emission scenario on flood indices was studied by Osuch et al. (2016) whilst the low flows were analysed by Osuch et al (2017). Both those studies indicated that emission scenario choice has a relatively small influence on the results. The implication of the choice of only one emission scenario will be explained in the revision.

RC 8. P5, L9: Why is QM applied in this study? The reasoning behind this choice is not completely clear from the preceding sentences.

AC 8. Many popular existing bias correction methods have been reviewed and compared and quantile mapping (QM) was found to outperform other methods (Gudmundsson et al., 2012; Teutschbein and Seibert, 2013; Chen et al., 2013; Osuch et al., 2016). More recently, the standard non-parametric QM method has been adapted to more explicitly preserve the raw modelled climate change signals (Willems and Vrac, 2011; Sunyer et al., 2014; Cannon et al., 2015). This means, in the QM method, that a raw modelled value is always corrected by the same value of bias or error that is determined by its respective quantile in the reference period.

RC 9. P5, L18-19: Did Osuch et al. (2015) model the same catchment as in this study and therefore, can it be assumed that the same five parameters are sensitive? And are the same five parameters sensitive for low flows and for high flows? That would be remarkable.

AC9. The HBV model was applied in different hydro-climatic condition in Poland by different researchers and they found the five most sensitive parameters for both high flow and low flow characteristics. The set of five parameters chosen in this study was dictated by the most common catchment conditions. Therefore it is not surprising that

the same parameters are sensitive in both high and low flow conditions. However, in this study we used two objective functions to encapsulate the high and low flow characteristics instead of selected best parameters only belonging to low flow and high flow.

RC10. P6, L15-16: How many Monte Carlo simulations have been executed and is this number sufficient (compare with literature)?

AC 10. 20000 MC simulations were executed. Many research papers recommend above 10, 000 MC (e.g. Xiaoli Jin et al., 2010; Romanowicz et al., 2013; Houska T. et al., 2014).

RC 11. P6, L22: Is it common practice to determine the thresholds in an iterative way? The determination of the threshold based on the requirement that 95% of the observations should be in the 95% confidence interval seems to be reasonable. However, please refer to other studies employing the same approach.

AC 11. To our knowledge, it is a common practice. The thresholds determine the variance of the predictions. Too high a threshold results in too narrow confidence bands. By iteration we meant the "trial and error approach" which does not involve any algorithm. We would be surprised if the iterative determination of threshold values has not yet been introduced, but we are not aware of any studies that have followed this approach. We will change the wording to avoid confusion.

RC 12. P7, L4-5: In general it is doubtful whether distributions with a 'large' number of parameters will model data in a more accurate way than distributions with a small number. This partly depends on the data quantity and quality and similarly as in hydrological modelling there will be a balance between the complexity of the distribution (i.e. number of parameters) and the amount of data (and quality).

AC 12. We agree with the reviewer that there must be a balance achieved between the complexity of the distribution (i.e. the number of parameters) and the quality of data.

We admit that this sentence can be deleted as it is a too large generalization.

RC 13. P7, L7-8: What does an 'overall good performance' mean? Compared to which other distributions?

AC 13. A number of distributions was tested including a three-parameter lognormal and an inverse Gaussian. GEV was the only distribution that performed well both for the high and low flow extremes. Although it was not necessary to use the same distribution for both extremes, it made our discussion more transparent.

RC 14. P7, L25-27: It is not completely clear why the analyses are performed for a period of 130 years. Since the manuscript is about impacts of climate change on hydrological extremes, you would expect a comparison between historic and future climate conditions. Furthermore, climate change automatically implies the existence of nonstationarity and as such, by considering a period of 130 years assuming stationarity by using the same extreme value distribution will result in serious flaws.

AC 14. We do not think that the impact of climate change on hydrological extremes should be based on a comparison between historic and future climate conditions. What we propose here is to study the trend of projected indices instead of the "change". The Biala Tarnowska catchment does not show any non-stationarity in the extreme flow events (Meresa et al. 2017, submitted for publication). Therefore it is a suitable catchment to compare both approaches. We are aware that non-stationary flood frequency analysis has to be applied for non-stationary extreme events. We want to show here that taking 30-year long time-series to compare between reference and future periods involves large uncertainty even for 30-year return flows. The uncertainty ranges of 30-year return period flows obtained using 130-year long time series can be nearly four times smaller.

RC 15. P8, L7-12: The idea behind this section is not clear. Why is the trend in daily annual maximum precipitation and temperature analysed while the interest is in uncertainty in hydrological indices with climate change? Moreover, why is the daily

annual maximum precipitation of interest and not for instance the two-day or three day precipitation (which might be stronger correlated to annual maximum discharge values)? Which temporal resolutions of precipitation are relevant for annual minimum flows? And what is the supposed role of daily annual maximum

AC 15. The idea behind presenting the precipitation and temperature patterns was to show the variability of driving forces behind the changes in flow extreme index. However, the idea was not properly explained and followed. For a catchment of that size, daily maximum and mean sums of precipitation are well correlated with the flow patterns. The temperature patterns, on the other hand, present the changes in the evaporation losses and possibly, indicate the changes in flood regime.

RC 16. P8, L14-20: How have the different criteria for high and low flows been applied in continuous hydrological modelling for periods of 30 years (or more)? When is the 'high flow' parameter set being used and when the 'low flow' one? What is the threshold for low flows and high flows; a specific discharge value or exceedance frequency?

AC 16. As explained in section 3.3, we applied a stochastic formulation to the estimation of the HBV model parameters. That means that 20000 simulations of the HBV model were run for the 30-year long calibration period with parameters sampled randomly within the assumed parameter ranges. The calibration is performed using the observed precipitation, temperature and flow records. We applied logNSE criterion for low flow and NSE criterion for high flow index to all the simulated flow series. Then we evaluated thresholds for the criteria, called likelihood thresholds, based on the requirement of 95% of the observations should be in the 95% confidence interval separately for high and low flows (Table 3). In other words, we do not have one "high" or "low" parameter set but we have two multiple sets (each including thousands of parameter sets) representing "high" and "low" flow models.

RC 17. P9, L5: Which best parameter sets are meant here? When is the best low flow parameter set used and when the best high flow parameter set?

AC 17. Results shown in Fig. 4 were obtained from the HBV model simulations fed by the precipitation and temperature projections obtained from the seven GCM/RCM models under the RCP4.5 scenario for the best parameter sets from the MC parameter samples, giving the highest weights derived from the NSE for the high flows and logNSE for the low flows, respectively. The raw hydro-meteorological projections were applied to study the high flow index whilst bias corrected precipitation and temperature data were used for the low-flow index studies.

RC 18. P9, L7-8: 'twice as large'; where do we see that?

AC 18. Sentence will be corrected to: Obtained flow projections shown in Fig. 4, follow the rainfall projections shown in Fig. 2, with annual maximum flow values even four times larger than historical events occurring after 2016 for some GCM/RCM model projections.

RC 19. P9, L12-22: This evaluation is not clear to me. Why do the authors evaluate results at a monthly scale? How can you assess annual maximum flows for each month? What do the authors mean with 'range' of annual maximum flows?

AC 19. Thank you, it is corrected in the main manuscript, "annual" was replaced by "monthly". Analysis of variability of monthly flows was presented to illustrate seasonal changes of extreme flows in the near future period and the uncertainty related exclusively to hydrological model uncertainty for each climate model projection. Some changes of seasonality are visible for high flows, but low flows do not show any distinctive differences between reference period and near future. We agree with the reviewer that this section is not adding much to the paper scope and we will delete it, together with Fig. 5.

RC 20. P10, L9-10: The decrease in the spread of Q30 in the far future compared to the near future is strange. The authors should reflect on this. Is it related to the fact that only one RCP scenario is taken into account?

AC 20. This smaller spread of the far-future projected changes was also observed in the other climate impact studies on the same catchment (Osuch et al., 2017) for both the RCP4.5 and RCP8.5 emission scenarios using the HBV model for hydrological simulations. Research is on-going to explain that phenomenon.

RC 21. P10, L20-22: Also this observation needs discussion. Why the spread is more evenly distributed for minimum flows compared to maximum flows?

AC 21. This is related to the influence of the climate model spread on the simulations (Osuch et al., 2017). It is much bigger for high flows and not very big for the low flows. We also have to remember that low flow simulations used bias-corrected meteorological drivers whilst the high-flow simulations were driven by the raw data. Bias correction decreases the variability of climate models.

RC 22. P11, L13-14: Are the relative differences for annual minimum flows also smaller?

AC 22. Yes. The sentence should read: The relative differences obtained for the annual minimum flow Q30 estimates are smaller, suggesting that low flow quantiles are less susceptible to the errors related to the length of the evaluation period.

RC 23. P12, L7-9: This is an interesting topic, but has not been investigated in this study since only one catchment has been considered.

AC 24. We agree with the reviewer, this sentence is out of context and should be deleted.

RC 24. P12, L11-14: This is an interesting result assuming that all methodological steps are logical and correctly carried out. What is the reason for the importance of uncertainty due to climate models for high flow and the important of hydrological model parameter uncertainty for low flows? This is very important and interesting to discuss.

AC 24. The important role of hydrological model uncertainty in low flow predictions was already noticed in forecasting (Beninga et al., 2017). That effect can be explained

by the ratio of the prediction noise (in this case described by the hydrological model uncertainty) to the input signal which is much higher for low flows.

RC 25. P12, L23-24: What do the authors mean with 'this allows the problem of nonstationarity of model parameters to be avoided'?

AC 25. The sentence should read: (iii) Conditioning of the hydrological model was performed using different criteria for low and high flows in order to ensure the best model fit for the extremes; this does not solve the problem of non-stationarity of model parameters but allows for focusing on parameter sets adequate for low or high flow regimes.

RC 26. P12, L29-31: This statement seems to be obvious; the larger the ratio of return period vs. data length the higher the uncertainty. However, this extrapolation uncertainty is not explicitly assessed in this manuscript.

AC 26. Thank you for the comment. This statement should read "analysis of the influence of length of time series records on the uncertainty bands of the high flow quantile estimates and their changes suggests that the ranges of quantiles of return periods Q30 are up to four times smaller when the long-term flow projections are used (Table 4). The low flow Q30 quantiles are less influenced by the length of the record.

RC 27. P23, Table 2: The ranges defined by the lower and upper bounds frequently do not match with the optimal values (e.g. for ALFA, PERC, CLFUX). Can you explain this? Furthermore, some lower and upper bounds are exactly the same. Does this indicate that these parameters are deterministic? What about CFMAX (not mentioned as sensitive in section 3.3)? Finally, an upper bound of 2 for LP is impossible and an optimum value of 1 is remarkable at least (it would mean only potential evapotranspiration under fully saturated conditions).

AC 27. We thank the reviewer for this comment. There was a mistake in Table 2. The HBV model was calibrated using GLUE and optimal parameter sets were derived in

the form of multiple parameter sets, different for the high and low flows. When applying this method there is no unique parameter set chosen, but instead, a multiple set of parameters, each with a weight corresponding to the model performance criterion, represents the solution of a calibration problem. Therefore, there is no 'optimal" single solution to the calibration problem, even though a solution with the best goodness of fit criterion can be specified. Therefore this table should not include the "optimal" parameter values. The corrected Table 2 is at the end of the responses.

Technical corrections

RC 1. P1, L11: What is the distribution fit?

AC 1. 'distribution fit' will be changed into "theoretical distribution fit error"

RC 2. P1, L13: What kind of weighting do the authors mean?

AC 2. "with a separate criterion for high and low flow extremes"

RC 3. P1, L16: What is the difference between climate model variability and climate projection ensemble spread? Please use a consistent terminology.

AC 3. The meaning of "variability" is not the same as "spread". Here we meant "variability".

RC 4. P2, L3: What is inverse modelling in this respect? Is this term commonly used for calibration and validation purposes based on observed (historic) data?

AC 4. "Conditioning" can be used here instead of "inverse modelling", if it is clearer. Inverse modelling refers to model parameter calibration based on historical data.

RC 5. P2, L6: "weighting" instead of "weighing".

AC5. Corrected: "weighting" instead of "weighing".

RC 6. P3, L8: What is the 'relevant variability' of extreme index estimates?

AC 6. Changed into "a direct assessment of variability of extreme index estimates"

RC 7. P3, L19: The case study has already been mentioned.

AC 7. Thank you, the sentence will be deleted.

RC 8. P3, L30: The maximum daily precipitation? During which period?

AC 8. Thank you, corrected. 'Maximum precipitation was 68.3 mm d-1 and annual mean streamflow is 0.4 m3s-1 over the observation period.' Changed to 'The annual maximum precipitation, annual minimum streamflow and annual mean streamflow of the catchment were 68.3 mm, 0.4 m3s-1 and 5.43 m3s-1 respectively over the observation period (1971-2000)'.

RC 9. P3, L30-31: Which period for the streamflow) Isn't 0.4 m3/s a very low value for catchment area of about 1000 km2?

AC 9. Thank you. It is the same as with the previous comment. It is a minimum streamflow.

RC 10. P4, L12-14: Why do the authors use these complex abbreviations for the GCM/RCM combinations? It is not clear what the meaning of all the numbers is. Try to be consistent with the descriptions in Table 1.

AC 10. Corrected as in the Table 1 included at the end of these responses.

RC 11. P5, L12: Do you have a reference for the Matlab version of HBV?

AC 11. The MATLAB version of HBV used in this study was based on Lindstrom et al (1997). The original MATLAB code from Twente University NL, was further developed and adjusted for the purpose of climate impact studies in the Institute of Geophysics PAS.

RC 12. P5, L15-17: Only 12 out of 14 HBV parameters are mentioned. In which routines can we find CFLUX and PERC (see line 19)?

AC 12. Thank you. It is changed to 'These routines are governed mainly by fourteen
HBV parameters, of which, six (TT, TTI, CFMAX, DTTM, CFR, WHC), three (FC, LP, BETA, CFLUX), two (KF, ALPHA) and one (KS, PERC) parameters are representing each routine respectively.'

RC 13. P5, L17: 'routines' instead of 'routing stage'?

AC 13. Thank you; corrected: 'routines' instead of 'routing stage'?

RC 14. P6, L24-P7, L3: This general description of the GEV distribution is not necessary here and can be found in many text books.

AC 14. It will be deleted

RC 15. P7, L16-17: What do the authors mean with ": : : aggregated speared of flow quantile change : : :"?

AC 15. "...aggregated speared of flow quantile change ..." meant "integrated spread ...

RC 16. P7, L19: 'squared' instead of 'squere'.

AC 16. Thank you, it is corrected in the main manuscript.

RC 17. P7, L22: The title suggests that the results of this study will be described. Please rephrase the title.

AC 17. "Description of the results" would be better?

RC 18. P7, L23: Different temporal resolutions? Shouldn't it be different lengths of data periods?

AC 18. Agree: 'different temporal resolutions' will be changed to 'different lengths of data periods'

RC 19. P7, L18: The meaning of all variables should be explained in the text.

AC 19. All the variables will be explained: where: Where: T_SSijk is total sum square

error for the specific hydrological extreme indicator (e.g. relative change in Q30) for the ith parameter sets range, jth climate model and kth distribution parameter range and $\mu$ is the overall mean and $\varepsilon\_ijk$ denotes the white Gaussian error.
RC 20. P8, L6: "Results and discussion"?

AC 20 Thank you, it is corrected in the main manuscript. As "Discussion of the results"

RC 21. P9, L2: 'the 10-year moving average from the ensemble mean'?

AC 21. Thank you, it is corrected in the main manuscript. Corrected as 'the 10-year moving average from the ensemble mean' changed to 'mean from the ensemble of seven climate models'

RC 22. P9, L15-16: Fig. 5a is mentioned twice.

AC 22. Fig. 5 will be deleted

RC 23. P9, L29-30: Decreases in minimum flows and increases in maximum flows? Shouldn't it be the other way around (according to the caption of Fig. 6)?

AC 23. Thank you, it is corrected in the main manuscript: 'Figure 6. Empirical flow quantiles of annual maximum flow (upper panels) and annual minimum flow (lower panels) under baseline and future climates (near and far future periods); the climate model spread is presented as a shaded area; green line denotes the mean value from all the GCM/RCM model realizations, red line denotes the averaged results obtained for the reference period.'

RC 24. P10, L6-7: Here, the annual minimum flows increase (see previous comment).

AC 24. Thank you. It is corrected as previous comment

RC 25. P10, L9: What is Q30? Commonly, that is a discharge with a non-exceedance frequency of 30%. However, here it seems to be an annual maximum flow with a return period of 30 years?

[Figure]

AC 25. Yes, it is annual maximum flow with a return period of 30 years. For annual maximum flow those two definitions have the same meaning. However, now to avoid confusions, we used as Qt30 in the main manuscript.

RC 26. P11, L7: 'Table 4' instead of 'Table 3'.

AC 26. Thank you, it is corrected in the main manuscript. 'Table 4' instead of 'Table 3'.

RC 27. P11, L26-P12, L2: The first part of the conclusions can be omitted (can be part of introduction section).

AC 27. Thank you, it is corrected in the main manuscript. Deleted

RC 28. P12, L9: 'hydrological parameter uncertainties' instead of 'hydrological model uncertainties'?

AC 28. Thank you, it is corrected in the main manuscript. Corrected to 'hydrological parameter uncertainties' instead of 'hydrological model uncertainties'

RC 29. P12, L24-27: This is a repetition of lines 11-14.

AC 29. Thank you, it is corrected in the main manuscript. Deleted

RC 30. P13, L3: A paper in preparation should not be included in the reference list.

AC 30. This paper has already been submitted (see references included). RC 31. P13-17: The reference list and referencing contain many errors, typos and inconsistencies. This should be carefully and thoroughly double-checked.

AC 31. The reference list will be corrected.

RC 32. P18, Fig. 1: What is the unit of the DEM map?

AC 32. The unit is meter. Thank you, it is corrected in the main manuscript.

RC 33. P18, Fig. 2: The interquantile range of what? Of the seven GCM-RCM combinations? In that case it would be better to show the individual model results, i.e. one

annual maximum for each combination so 7 points per year.

AC 33. Following the reviewer's advice raw projections of annual maximum precipitation and temperature for seven GCM/RCM combinations will be presented in Fig.2.

RC 34. P19, Fig. 3: In particular the scale of the upper panel looks strange. Flows in cubic mm? How accurate is your model? Please use the same (realistic) x-axis ranges.

AC 34. We guess that the reviewer means Fig. 4. The y-axis units should be in cubic meters per second. The figure y-axis will be corrected.

RC 35. P19, Fig. 4: This figure (and also Fig. 2) is too small. What do we see here?

AC 35. Fig. 4 presents projected annual maximum and minimum flow. The figures 2 and 4 will be enlarged.

RC 36. P20: The differences between historic and future periods cannot be clearly seen in these figures.

AC 36. Following the reviewer's comments we decided to delete Fig.5 together with the subsection 4.4.

RC 37. P21, Fig. 6: What are the different lines in these figures? And is baseline and reference period the same?

AC 37. Thank you, it is corrected in the main manuscript. Changed to 'Figure 6. Empirical flow quantiles of annual maximum flow (upper panels) and annual minimum flow (lower panels) for the reference period and future climates (near and far future); the climate model spread is presented as a shaded area; green line denotes the mean value from all the GCM/RCM model realizations in each period (near and far future period), red line denotes the averaged results obtained for the reference period. Each black line represents individual climate models'

RC 38. P21, Fig. 7: In the caption 'right hand panel' is mentioned twice.

AC 38. The figure caption will be changed to: Total uncertainty ranges of theoretical GEV-based annual maximum (left hand panels) and minimum (right hand panels) flow quantiles over 30 year periods for the Biala Tarnowska at Koszyce; upper panels - the reference period 1971-2000; middle panels - near future 2021-2050; lower panels - far future 2071-2100. RC 39. P22, Fig. 8: Idem, annual minimum flow is mentioned twice.

AC 39. Changed to 'annual maximum flow as a function of return level (left panel panel)'

RC 40. P23, Table 1: Which meteorological institute is connected to RACMO?

AC 40. Netherlands.

RC 41. P23, Table 2: The caption is not clear.

AC 41. Changed by 'Table 2. HBV parameter ranges: upper band (UB), lower band (LB) value' (Table 2 enclosed).

RC 42. P24, Table 4: What do the authors mean with 'change in width of ...'? What compared to what?

AC 42. Change in the 95% confidence interval width

References Benninga, J-H., Booij, M., Romanowicz, R.J., Rientjes, T.H.M.: Performance of ensemble streamflow forecasts under varied hydrometeorological conditions, Hydrol. Earth Syst. Sci. Discuss., doi:10.5194/hess-2016-584, 2016. Blasone, R., Vrugt, J., Madsen, H., Rosbjerg, D., Robinson, B., & Zyvoloski, G.: Generalized likelihood uncertainty estimation (GLUE) using adaptive Markov Chain Monte Carlo Sampling. Advances in Water Resources (31), 630-648. 2008. Cannon, A. J., Sobie, S. R., and Murdock, T. Q.: Bias correction of GCM precipitation by quantile mapping: How well do methods preserve changes in quantiles and extremes?, J. Climate, 28, 6938-6959, doi:10.1175/JCLI-D-14-00754.1, 2015. Clarke, L., Edmonds, J., Jacoby, H., Pitcher, H., Reilly, J. and Richels, R.: CCSP Synthesis and Assessment Product 2.1, Part A: Scenarios of Greenhouse Gas Emissions and Atmospheric Concentrations. U.S. Government Printing Office. Washington, DC, 2007. Houska, T., Multsch, S., Kraft, P., Frede, H.-G. and Breuer, L.: Monte Carlo-based calibration and uncertainty analysis of a coupled plant growth and hydrological model. Biogeosciences, 11, 2069-2082, doi:10.5194/bg-11-2069-2014, 2014. Meresa, H., Romanowicz, R.J., Napiorkowski, J.J.: Trends of projections of hydrological extreme indices in the 21st century, Acta Geophysica, submitted, for publication, 2017. Osuch, M., Romanowicz, R.J., Wong, W.: Analysis of low flow indices under varying climatic conditions in Poland, Hydrology Research, submitted for publication, 2017.

Smith, SJ., and Wigley, TML.: Multi-Gas Forcing Stabilization with the MiniCAM. Energy Journal SI3:373-391, 2006. Strupczewski, W., Kochanek, K., Markiewicz, I., Bogdanowicz, E., Weglarczyk, S. and Vijay P. Singh, PV.: On the tails of distributions of annual peak flow. IWA Publishing 2011 Hydrology Research 9 42.2-3, 9, 2011. Sunyer, M.A., Hundecha, Y., Lawrence, D., Madsen, H., Willems, P., Martinkova, M. Vormoor, K., Burger, G., Hanel, M., Kriauciuniene, J., Loukas, A., Osuch, M., and Yucel, I.: Inter-comparison of statistical downscaling methods for projection of extreme precipitation in Europe, Hydrol. Earth Syst. Sci., 19, 1827-1847, doi:10.5194/hess-19-1827-2014, 2014. Willems, P., and Vrac, M.: Statistical precipitation downscaling for small-scale hydrological impact investigations of climate change, J. Hydrol., 402, 193-205, doi:10.1016/j.jhydrol.2011.02.030, 2011. Wise, M., Calvin, K., Thomson, A., Clarke, L., Sands, R., Smith, SJ., Janetos, A., Edmonds, J.: Implications of Limiting $CO_2$ Concentrations for Land Use and Energy. Science 324:1183-1186, 2009. Xiaoli, Jin., Xu, CY., Qi Zhang, Qi. and Singh, VP.: Parameter and modeling uncertainty simulated by GLUE and a formal Bayesian method for a conceptual hydrological model. Journal of Hydrology 383, 147-155, 2010.

| GCM | RCM | expansion name | Institute |
|---|---|---|---|
| EC-EARTH | RCA4 | Rossby Center regional | Swedish Meteorological and Hydrological Institute |
| EC-EARTH | HIRHAM5 | Atmospheric model | Danish Meteorological Institute |
| EC-EARTH | CCLM-4-8-17 | Community land model | NCAR UCAR |
| EC-EARTH | RACMO22E | Regional atmospheric climate model | Meteorological institute |
| MPI-ESM-LR | CCLM4-8-17 | Community land model | Max Planck Institute for Meteorology |
| MPI-ESM-LR | RCA4 | Regional-scale model | Max Planck Institute for Meteorology |
| CNRM-CM5 | CCLM4-8-17 | Community land model | CERFACS, France |

**Fig. 1.** Table 1 List of GCM/RCM models used in this study

| Parameter | Description | LB | UB | Unit |
|-----------|-------------|-----|-----|------|
| FC | Maximum soil storage | 0.1 | 250 | mm |
| BETA | Shape coefficient | 0.01 | 7 | - |
| LP | SM threshold for reduction of evaporation | 0.1 | 1 | - |
| ALFA | measure for non-linearity of flow in quick runoff | 0.2255 | 0.2255 | - |
| KF | recession coefficient for runoff from quick runoff | 0.2826 | 0.2826 | $d^{-1}$ |
| KS | recession coefficient for runoff from base flow | 0.0005 | 0.3 | $d^{-1}$ |
| PERC | percolation rate occurring when water is available | 0.01 | 100 | $mm\ d^{-1}$ |
| CFLUX | Rate of capillary rise | 1.0003 | 1.003 | $mm\ d^{-1}$ |
| TT | Threshold temperature for snowfall | 1.0145 | 1.0145 | $^0C$ |
| TTI | Threshold temperature interval length | 7 | 7 | $^0C$ |
| CFMAX | Degree day factor | 0 | 20 | $mm\ ^0C^{-1}\ d^{-1}$ |
| FOCFMAX | Rate of snowmelt | 0.1484 | 0.1484 | $mm\ ^0C^{-1}\ d^{-1}$ |
| CFR | Refreezing factor | 0.2779 | 0.2779 | - |
| WHC | Water holding capacity of snow | 0.001 | 0.001 | $mm\ mm^{-1}$ |

**Fig. 2.** Table 2. HBV parameter ranges: lower band (LB), upper band (UB), unit

---

## Author Comment (AC4) · 13 Mar 2017

Dear Editor and Reviewer,

This is the corrected version of the response to the reviewer comments. Please ignore the previous version which was not properly formatted (the lines merged). We also added the tables 1 and 2.

Thank you very much for the constructive comments that will help to considerably improve and clarify the manuscript. All comments have been addressed point-by-point. Following the reviewer' feedback we will make the corresponding changes in the manuscript.

Anonymous Referee #2

Overview

RC. The authors assess the effect of different uncertainty sources on climate change projections. The presentation of the results is easy to follow and interpret. Especially Figure 9 is very informative. However, there is room for improvement using specific comments and checklist below. I recommend major revision as the model calibration part is not clear.

AC. We thank the reviewer for concise and valuable comments.

Specific Comments:

RC1. Table 2: Optimal values of some parameters are out of lowerand upper limits e.g. CFMAX which cannot be reached by an algorithm e.g. SCEUA, CMAES etc. How was this achieved by a calibration algorithm? Did you follow a manual calibration scheme?

AC1.The table 2 is now corrected (included at the end of this file). We do not use deterministic calibration, instead the GLUE -based stochastic calibration is applied. When applying this method there is no unique parameter set chosen, but instead, a multiple set of parameters, each with a weight corresponding to the model performance criterion, represents the solution of a calibration problem. Therefore, there is no 'optimal" single solution to the calibration problem, even though a solution with the best goodness of fit criterion can be specified.

RC2. Demirel et al (2013a) is in the reference list but not in the text.

AC2. Thank you, it has been corrected.

RC3. Please explain the abbreviations used at legend in figure caption. The legend of Fig8 is confusing: "distn"?

AC3. Thank you, it has been corrected. "distn" replaced by "distribution"

RC4. Did you compare uncertainty in HBV model parameters with other studies (Addor et al., 2014; Demirel et al., 2013b; Osuch et al., 2015) using HBV model for forecasting hydrological extremes? How would the results overlap for 10 day forecast (Demirel et al., 2013b) and long term climate predictions in EUROCORDEX (dataset used in this study)?

AC4. The uncertainty in the HBV model parameters was compared with the other studies, including Osuch (2015) and Demirel et al (2013b). Demirel et al. (2013b) explored the influence of uncertainty in input, hydrological model parameters and initial conditions on a 10-day ensemble flow forecasts. The results showed that parameter uncertainty had the largest effect on the medium range low flow forecasts, which is consistent with the present paper findings. Addor et al. (2014) concentrated on the influence of different hydrological model structure, involving three hydrological models, emission scenarios, climate models, post-processing and catchments. Their results indicate that influence of model structure varies with the catchment. However the authors did not take into account hydrological model parameter uncertainty, which is the main focus of the present paper. Osuch (2015) compared three sensitivity analysis techniques to describe the HBV model parameter interactions. We used the output of that paper to eliminated less sensitive HBV model parameters in order to minimize computational cost.

RC5. Fig5: Parameter uncertainty should be presented differently to assess the contribution of each parameter uncertainty to total uncertainty. From this figure the reader can't see the most uncertain parameter. A figure similar to Figure 4 in Demirel et al (2013b) or Fig9 in the current manuscript can be very useful for modelers. This can be easily done as the GLUE results would allow such ranking.

AC5. Thank you for the comment. We decided to delete this figure and subsection 4.4 following the first reviewer comments.

RC6. Conclusion 2 (ii): Please explain the drizzle effect? Not clear.

AC6. Simulated climate variables (precipitation and temperature) by individual GCMs/RCMs often do not reach agreement with observed climate time series. This is due to the effect of systematic and random model errors of GCMs/RCM simulations. Such systematic errors lead to simulate many drizzle days (i.e., too many days with very low precipitation intensity and too few dry days). The drizzle effect is related to the performance of climate models. It presents itself in the form of frequent rainfall of a very small intensity. The physics behind precipitation generation is very complex and involves processes operating on a wide range of scales. The frequent 'drizzle' is produced mainly by convective parameterization. It appears in many climate models and invokes errors in the intensity and frequency of precipitation (Terai et al. 2016). The correction can be performed using the number of wet days in a month (Osuch et al. 2016). Because of this bias in precipitation, using direct climate model output as inputs to hydrological modelling for low flow analysis often leads to unrealistic results and therefore bias correction is required in the case of low flow projections.

RC7. Section 3.6 and Conclusion 5 (v): Is ANOVA method a global or local sensitivity analysis method? Can interactions (parameter etc.) be assessed using this method? Why ANOVA is used instead of other elementary and global methods e.g. Morris, SOBOL, PEST, FAST etc. These aspects of the ANOVA method should be described in section 3.6 and conclusions should follow these details.

AC7. Nowadays, many global sensitivity methods have been proposed and used, such as Fourier amplitude sensitivity test (FAST), Regional Sensitivity Analysis (RSA), Analysis of Variance (ANOVA), Parameter Estimation Software (PEST), Morris, and Sobol method. Among these global sensitivity analysis methods, ANOVA is proved to be one of the most robust and effective tools to analyze both continuous and discrete factors (Montgomery, 1997), and it is widely applied in hydrology (Bosshard et al., 2013; Zhan, et al., 2013; Lagerwalla, et al., 2014; Addor et al.,2014; Giuntoli et al., 2015; Osuch, 2015). We used ANOVA approach due to its numerical facility (MATLAB) and ability to evaluate the main and interactive effects between factors considered.

RC8. Conclusion bullets are confusing. Two times "iv" exists and sentences are not clear. There are typos too. For example Conclusion vi should start with capital. Please rephrase them with short and clear conclusions. And relate them to the results section. Bullet conclusions in Demirel et al (2013b) can be an example. For each result section one paragraph is given in conclusion.

AC8. Thank you, it is corrected in the main manuscript.

References

Addor, N., Rossler, O., Koplin, N., Huss, M., Weingartner, R., and Seibert, J.: Robust changes and sources of uncertainty in the projected hydrological regimes of Swiss catchments, Water Resour. Res., 50, 7541-7562, doi:10.1002/2014WR015549, 2014.

Bosshard, T. Carambia, M. Goergen, K. Kotlarski, S. Krahe, P. Zappa, M. and Schar, C.: Quantifying uncertainty sources in an ensemble of hydrological climate- impact projections, Water Resour. Res, 49, 1523-1536, doi:10.1029/2011WR011533, 2013.

Demirel, M. C., Booij, M. J. and Hoekstra, A.Y.: Effect of different uncertainty sources on the skill of 10 day ensemble low flow forecasts for two hydrological models, Water Resour. Res., 49, 4035–4053, doi:10.1002/wrcr.20294, 2013b.

Giuntoli, J., Vidal, J.-P., Prudhomme, C. and Hannah, D.M.: Future hydrological extremes: the uncertainty from multiple global climate and global hydrological models, Earth Syst. Dynam., 6, 267-285, 2015. doi:10.5194/esd-6-267-2015

Lagerwalla, G.; Kiker, G.; Muñoz-Carpena, R.; Wang, N.: Global uncertainty and sensitivity analysis of a spatially distributed ecological model. Ecol. Model. 2014, 275, 22-30.

Osuch M.: Sensitivity and uncertainty analysis of precipitation-runoff models for the middle Vistula basin, in GeoPlanet: Earth and Planetary Sciences, DOI:10.1007/978-3-319-18854-6-5, 2015.

Montgomery, D.C.: Design and Analysis of Experiments. Wiley and Sons Ltd.: New York. 1997.

Zhan, Y., Zhang, M.: Application of a combined sensitivity analysis approach on a pesticide environmental risk indicator. Environ. Model. Softw., 49, 129-140, 2013.

Terai, CR. Caldwell, P. and Klein, S.A.: 2016. Why Do Climate Models Drizzle Too Much and What Impact Does This Have. Abstract A 53K-01, American Geophysical Union (AGU) fall meeting, San Francisco 12-16, 2016.

| GCM | RCM | expansion name | Institute |
|---|---|---|---|
| EC-EARTH | RCA4 | Rossby Center regional | Swedish Meteorological and Hydrological Institute |
| EC-EARTH | HIRHAM5 | Atmospheric model | Danish Meteorological Institute |
| EC-EARTH | CCLM-4-8-17 | Community land model | NCAR UCAR |
| EC-EARTH | RACMO22E | Regional atmospheric climate model | Meteorological institute |
| MPI-ESM-LR | CCLM4-8-17 | Community land model | Max Planck Institute for Meteorology |
| MPI-ESM-LR | RCA4 | Regional-scale model | Max Planck Institute for Meteorology |
| CNRM-CM5 | CCLM4-8-17 | Community land model | CERFACS, France |

**Fig. 1.** Table 1 List of GCM/RCM models used in this study

| Parameter | Description | LB | UB | Unit |
|-----------|-------------|-----|-----|------|
| FC | Maximum soil storage | 0.1 | 250 | mm |
| BETA | Shape coefficient | 0.01 | 7 | - |
| LP | SM threshold for reduction of evaporation | 0.1 | 1 | - |
| ALFA | measure for non-linearity of flow in quick runoff | 0.2255 | 0.2255 | - |
| KF | recession coefficient for runoff from quick runoff | 0.2826 | 0.2826 | $d^{-1}$ |
| KS | recession coefficient for runoff from base flow | 0.0005 | 0.3 | $d^{-1}$ |
| PERC | percolation rate occurring when water is available | 0.01 | 100 | $mm\ d^{-1}$ |
| CFLUX | Rate of capillary rise | 1.0003 | 1.003 | $mm\ d^{-1}$ |
| TT | Threshold temperature for snowfall | 1.0145 | 1.0145 | $^0C$ |
| TTI | Threshold temperature interval length | 7 | 7 | $^0C$ |
| CFMAX | Degree day factor | 0 | 20 | $mm\ ^0C^{-1}\ d^{-1}$ |
| FOCFMAX | Rate of snowmelt | 0.1484 | 0.1484 | $mm\ ^0C^{-1}\ d^{-1}$ |
| CFR | Refreezing factor | 0.2779 | 0.2779 | - |
| WHC | Water holding capacity of snow | 0.001 | 0.001 | $mm\ mm^{-1}$ |

**Fig. 2.** Table 2. HBV parameter ranges: lower band (LB), upper band (UB), unit

---

## Author Response (AR2)

Dear Editor and the reviewers,

We thank the reviewers and the associate editor for their relevant, valuable and constructive comments, which add value to our manuscript. We revised our manuscript following the reviewer's comments. Please find enclosed the revised manuscript, and a version containing all the changes to be visible. Each comment by the reviewer is first recalled, then the corresponding replies are given.

RC= Reviewer Comment;
AR= Author Response

General comments

I would like to thank the authors for the thorough revision of the paper and the response to my comments. The paper has substantially improved, although still some issues are not completely clear and sufficiently substantiated (e.g. the non-stationarity issue, see specific comment 14). Furthermore, the English writing style can be further improved. Spelling, style and grammatical errors can still be found in the manuscript. My specific comments and technical corrections are given below.

**Specific comments**

RC1. P1, L11: What is the 'theoretical distribution fit'? The fit of what? This is not clear and cannot be understood from the abstract alone.

AR1. Theoretical distributions are based on mathematical formulas in building a frequency curve. The sentence was changed to:
".... the third related to the error in fitting theoretical distribution models to annual extreme flow series."

RC2. P2, L1: Is only $CO_2$ included in this scenario? Greenhouse gas scenario seems to be more appropriate here.
AR2. Thank you, we agree with the review $CO_2$ is one type of greenhouse gases. So, it is changed from "$CO_2$" to "Greenhouse gas emission"

RC3. P2, L22-33: The structure of this part is not very clear. Several issues are discussed here and there is a sudden transition from the topic of uncertainty to bias correction methods in line 26. Please try to improve the structure of this part of the introduction.
AR3. We moved the downscaling issue to the previous paragraph.

RC4. P2, L28: "… are dealt with …"; in this study or in other studies?
AR4. Thank you, corrected in the main manuscript as "In a number of studies the hydrological model structural and parametric errors are dealt with using a multi-model approach….."

RC5. P2, L33: "… which is the main focus of the present paper …"; the main focus has not been introduced yet and moreover, besides hydrological model parameter uncertainty two other sources of uncertainty are the focus of this paper.
AR5. Thank you, it is corrected in the main manuscript by "….which is the main focus of the present paper…." changed to "…which is included in the present paper…."

RC6. P3, L9-12: The aim of the paper is still not very clear. The authors suggest that the influence of hydrological model uncertainty is assessed while only the influence of hydrological parameter uncertainty is assessed (and not for instance model structure). Furthermore, it seems the spread of extreme indices is related to the spread due to different climate models and the distribution fitting error (or theoretical distribution fit as in the abstract). However, while the main idea of the paper seems to be to compare the effects of three sources of uncertainty (parameters, climate models, distribution fit) on the uncertainty in hydrological extremes, the research aim suggests something different.

AR6. Thank you, we will consider it in the main manuscript as "In this study, we assess the critical role of uncertainty in the projection of future hydrological extremes in the Biala Tarnowska mountainous catchment in Poland in the 21st century. We consider three sources of uncertainty. These are epistemic climate projection uncertainty, hydrological model parameter uncertainty and uncertainty of extreme index estimates (the error in fitting theoretical distribution models to annual extreme flow series). We restricted the sources of epistemic uncertainty to that which have the largest impact – i.e. climate model spread, omitting the uncertainty related to bias correction and emission scenario. The error related to the distribution fit was included as an essential part of the extreme index evaluation, which requires extrapolation of annual maximum or minimum flow distributions to higher order quantiles (e.g. 1-in-100 year, or 1-in-200 year). "

RC7. P3, L13-19: I appreciate the explanation of the authors regarding the choice of the uncertainty sources, but would like to advise them to mainly connect this explanation to the previous work of Osuch et al. (2016) and not for instance mentioning that uncertainty related to 'geomorphology of the catchment' is not taken into account. I.e. this latter source seems to be very specific (and arbitrary) and might even be slightly incorporated in the hydrological model parameter uncertainty.

AR7.we agree with the review. It is corrected in the main manuscript and "geomorphology of the catchment" is deleted.

RC8. P5, L3-4: How can these references refer to the RCP scenarios, which have been developed after 2010?

AR8. Thank you it is corrected in the main manuscript

RC9. P5, L26-28: Although this statement seems to be logical and obvious, this might depend on the case study/ catchment area considered?

AR9. The sentence was changed to: "Their results showed the single gamma distribution mapping to be the one which produced the observed characteristics most accurately of all the techniques and regions studied."

RC10. P6, L7-9: It is not clear to me whether the appropriateness of bias correction has been investigated for the study area used in this paper or for other catchments. In the former case, more evidence needs to be provided that the different cases with/ without bias correction are appropriate and in the latter case the authors need be make clear that we can assume the same 'bias correction conditions' for their study area as compared to other study areas. The application of the hydrological model with climate data with and without bias correction is described in section 3.7, but already in this section (3.2) it is decided to only use one set of data (with or without bias correction).

AR10. We appreciate for the review to see the point, the last sentence was moved to the section 3.7 and additional discussion on the choice between raw and bias-corrected precipitation projections was given.

RC11. P6, L23: "… following those studies"; why can we assume the same sensitive parameters as in 'those studies'? This is still not clear. Moreover, it is still hard to imagine that the same set of six sensitive parameters has been found for low and high flows.

AR11. Thank you, an additional explanation was added: The studies mentioned show that the sensitivity of HBV model parameters vary depending on the catchment characteristics and time period of model evaluation. However, the parameters chosen were the most sensitive in the widest range of input variability. The other HBV model parameters show small influence on model output independently of the event type. The application of different evaluation criteria for low and high flow conditioning allows for treating the parameters according to their influence on the flows, i.e. takes into account model output sensitivity (Saltelli et al. 2006). Secondly, we wanted to minimize the computational cost.

RC12. P7, L5-6: How did you combine the low flow and high flow parameter sets? Or did you run the HBV model with each set separately? Then, how is the performance of the HBV model with low flow parameters for high flows (and vice versa)? When calibrated for low flows (high flows) does the model still give reasonable results for other flow conditions and does the model give the right results for the right reasons?

AR12. Yes, as explained further in the paper, we run HBV model with each set separately (20000 parameter sets). In the validation stage, two posterior distributions are used, one based on the logNSE objective function for low flow and the other one based on the NSE objective function for high flow. As presented in the figure 3, the results satisfy the conditions. The reviewer's comment may refer to a deterministic approach not followed in this paper.

RC13. P8, L7: Do the authors mean 'annual maximum and minimum daily flows'? Why do they use annual minimum daily flows as an indicator of low flows? Wouldn't a minimum value averaged over multiple days be more appropriate (e.g. 7 or 10 days)?

AR13. Yes, we meant annual maximum derived from daily flow and annual minimum flow derived from seven days moving average flow. The explanation is added in the text.

RC14. P9, L9-11: It seems to be contradictory to have stationarity in extreme flow events while at the same time the authors conclude that there are substantial differences (increases and decreases) between extreme flows for the reference and future periods (see Figure 5 and section 4.3). How should we see this? Is the comparison between uncertainty ranges derived using the two approaches a fair comparison?

AR14. We thank the reviewer for that comment. We agree that the comparison of flood frequency curves obtained from 30 and 130 year series of annual extremes is not possible. However what we compare is the uncertainty range of flow quantiles derived from 30 year and 130 year-long annual extreme series. Such comparison is not related to non-stationary condition of the extremes but rather to the number of events included in the frequency derivation. It is important to note that 30-year based quantiles are highly uncertain and show unrealistic changes which are not visible in long annual extreme series. Moreover, these changes in 30-year based quantiles are caused by inter-decadal variability and

they depend on the starting year of those 30-year periods. This might explain large differences in the estimates of future quantile changes obtained in a number of studies (Kundzewicz et al., 2017).

The text was added in page 12 (after line 5) and also in page 13, starting from line 26.

RC15. P9, L23-24: How did you combine the 10 000 MC samples for GEV with 20 000 parameter sets from GLUE? How arbitrary is the result of this uncertainty analysis? Did you repeat the uncertainty analyses and were the results in line with the first uncertainty assessment?

AR15. The explanation was added page 9, line 23-24 (section 3.7): "10000 MC normal samples of the GEV model parameter space were performed for each of the behavioural parameter sets (section 4.2)."

RC16. P10, L4-6: I would still recommend to plot the individual model results in this figure, seven model combinations/ points do not justify the use of a box plot. In the caption it should be 'annual mean temperature' instead of 'annual maximum daily temperature'.

AR16. Thank you. It is corrected in the main manuscript. Changed 'annual maximum daily temperature' to 'annual mean daily temperature'. However, we are not convinced that that change gives more clear picture of annual extremes.

RC17. P10, L9-10: Do you have evidence that the annual precipitation sum is an important driver of low flows or is for instance a six-month sum much more important?

AR17. Low flow patterns are affected by long-term precipitation which is reflected in annual precipitation sums, whilst the high flow events have a short time scale and correspond to precipitation maxima. We added more explanation in the text (page 10, lines 13-14).

RC18. P10, L10-12: The increase of the temporal variability with time is not very clear from Figure 2.

AR18. We changed this sentence to: "The results show a visible increase of the annual mean temperature and mean values of annual sums of precipitation and annual maxima.do not show visible changes with time."

RC19. P10, L26-27: What are the maximum logNSE values for the calibration and validation periods?

AR19. Text was added (pages 10-11, lines 31-2): The maximum logNSE value in calibration and validation period is 0.6212 and 0.6995 respectively. The maximum NSE value in calibration and validation is 0.7827 and 0.8128 respectively.

RC20. P11, L20-25: The changes in annual minimum flows are also relatively smaller than the changes in annual maximum flows (see Figure 5). This can be clearly seen from this figure since the range of quantile values for the reference period is approximately the same for low and high flows.

AR20. We agree with the reviewer that the changes in annual minimum flows are relatively small. The text was corrected.

RC 21. P12, L22: Which confidence interval is considered here, 95% or 90%? In the first case, the upper and lower boundaries should be 0.975 and 0.025 respectively.

AR21. These were 0.95 confidence bands, i.e. the first case. The text was corrected.

RC 22. P12, L28-31: How should we read Figure 7? Are the different colours overlapping or additive? And how should we interpret the different points?

AR 22. The colours in Fig. 7 are additive. The description was given in the text (page 13, lines 12-14) : "The red dotted lines denote the median of climate ensembles, black dotted lines denote the median of hydrological model parameter sets and the blue dotted lines denote the median related to the GEV distribution parameter fit."

**Technical corrections**

RC 1. P3, L18: "… hydrological model parameter uncertainty …" instead of "… hydrological model uncertainty …".

AR 1. Thank you corrected in the main manuscript. Changed "… hydrological model parameter uncertainty …" instead of "… hydrological model uncertainty …".

RC 2. P4, L7-8: "Study area …" instead of "Study areas …".

AR2. Thank you corrected in the main manuscript. Changed "Study area …" instead of "Study areas …".

RC 3. P4, L12: "… the river is characterized …" instead of "… the river characterized …".

AR3. Thank you corrected in the main manuscript. Changed "… the river is characterized …" instead of "… the river characterized …".

RC 4. P4, L14: "… areal precipitation …" instead of "… aerial precipitation …".

AR4. Thank you corrected in the main manuscript. Changed "… areal precipitation …" instead of "… aerial precipitation …".

RC 5. P6, L26: "… to eliminate …" instead of "… to eliminated …".

AR5. Thank you corrected in the main manuscript. Changed "… to eliminate …" instead of "… to eliminated …".

RC 6. P7-9: The description of the GLUE methodology is spread over three sections (3.3, 3.4 and 3.7) Try to structure this in a better way.

AR 6. We regret but we cannot see that spread. In the section 3.3 the GLUE method is announced without any explanation. Section 3.4 explains the approach. The section 7 presents the experimental design (GLUE is not mentioned).

RC 7. P9, l29-P10, L2: Please try to harmonize the terminology, such as 'variability' vs. 'patterns' and 'projections' vs. 'changes'.

AR7. We thank the reviewer for the comment. The text was reworded: In the following section, we present an analysis of the variability of maximum precipitation and temperature series on annual basis to see the correlation between the projected climate and hydrological extremes. The idea behind presenting the precipitation changes was to show their possible relation with the changes in flow extreme indices. For a catchment of that size, annual maximum and mean sums of precipitation are well correlated with the flow patterns when the rainfall-driven flood regime prevails. The temperature

changes, on the other hand, present the changes in the evaporation losses and possibly, indicate changes in the flood regime.

RC 8. P10, L14-20: This has already been described in the methods and can be moved/ integrated there.

AR8. Thank you, moved to methodology "As explained in the section 3.3, a stochastic formulation is applied to the estimation of the HBV model parameters. That means, 20000 simulations of the HBV model were run for the 30-year long calibration period (1971-2000) with six parameters sampled randomly within the assumed parameter ranges (Table 2)." Corrected in the main manuscript (page 9 lines 16-18).

RC 9. P13, L21: Please clarify 'medium range'.

AR 9. Thank you, it is corrected in the main manuscript as "medium range' replaced by "mean value'

RC 10. P14, L23-26: Is this conclusion for the 1971-2100 time series?
AR.10  yes. The sentence was clarified.

[revised manuscript text omitted]

---

## Author Response (AR3)

Dear Editor

We thank the Editor for his comments. Comments from the Editor are addressed below. Each comment by the Editor is first recalled (EC), then the authors' replies are given (AR).

EC1. P4L11: what is TTI, this is no standard HBV parameter
AR1. TTI stands for  temperature threshold interval length (°C). The description is given in Osuch et al. (2015). The reference to this paper is given in page 6, lines 26-27.
It is now added in Table 2.

EC2. P4L14: Seibert & McDonnell: McDonnell is spelled with two L. However, the more suitable reference with regard to HBV applications in many differnt climates might be Seibert & Vis, 2016 (Seibert, J. and Vis, M.J., 2016. How informative are stream level observations in different geographic regions? Hydrological Processes, 30: 2498–2508, doi: 10.1002/hyp.10887). Of course, I leave the decision which one, and if at all, reference you use, fully to you.
AR2. Thank you, the reference is change in the main manuscript from "Seibert & McDonnell to Seibert and Vis"

EC3. Equations: please avoid using multiletter variable names, see also author instructions
AR3. Thank you, the equations style is changed in the main manuscript:

(.....to the parameter sets (P), climate models (C) and parameter distribution sets (D), from the spread of flow quantile change in the near and far future, we use the following ANOVA model:
$$T_{ijk} = \mu + P_i + C_j + D_k + (P + C)_{ij} + (P + D)_{ik} + (C + D)_{jk} + \varepsilon_{ijk} \qquad .....)$$
Where T is a ….

Following this change also the legend of Fig. 8 and Fig. 8 captions were changed:

Figure 8. Total variance in estimates for the percentage change in QT30 in 2021-2050 relative to the 1971-2000 reference period. Each colour represents the relative contribution of uncertainty in percent; C denotes climate model; D – distribution fit; P – hydrological model parameters; ERROR denotes the Gaussian error (Eq. 3); a "star" denotes the correlation between the factors (C, D and P).

EC4. Terai et al is a reference to an AGU presentation, the drizzle issue is well-known and I would think that there are more suitable references.
AR4. Thank you, one more reference is added in the main manuscript.
"Maraun, D.: Bias correction, quantile mapping, and downscaling: Revisiting the inflation issue. J. Climate, 26, 2137-2143, doi:10.1175/JCLI-D-12-00821.1, 2013."

EC5. Fig 2: figure caption: change "represents the median of seven 10 climate models" to "represents the median of the seven 10 GCM/RCM combinations", please also consider adding a legend in the figure
AR5. Thank you, the caption was changed:
„each coloured dot represents an individual climate model and black line represents the median of the seven GCM/RCM combinations."
We would prefer not to introduce the legend, as there is no space left in the Figure.

EC6. Fig 5&6, please change colors (for the red-green blind)
AR6. Thank you, the green colour was changed to blue in both figures.

EC7. Tab 1: the Expansion name refers to the RCM, I assume, but then two different names are used for RCA4, pls check and correct

AR7. Thank you, the expansion name of RCM is corrected in the main manuscript.
from "Rossby center regional" to "Regional-scale model"

EC8. Tab 2: These parameter names do not fully fit to the text on P4, e.g. here is no TTI and on P4 is no FOCFMAX (what is this parameter?). Also, your values for WHC are unusual small and those for the refreezing coefficient unusual high. Please motivate these values! The lower limits of BETA and FC also sound unusual small.

AR8. Thank you for the comment. In Table 2 we added TTI and changed the name FOCFMAX to DTTM to keep it consistent with the text. In the choice of the ranges of parameter values we followed the study of Osuch et al. (2015), as explained in the text (p. 6, lines 24-32). WHC and CFR do not have large influence on the simulated flows and were chosen following the results of the HBV model calibration performed for this particular catchment.

[revised manuscript text omitted]

---

## Author Response (AR4)

Editor Decision: Publish subject to minor revisions (further review by Editor) (26 Jun 2017) by Jan Seibert

Comments to the Author:

Dear authors,

Thanks for revising your manuscript. However, there are still a few issues:

Equations 1 & 2 still are not correct mathematically (multiletter variables!)

Please check the references carefully; there are errors (e.g. Lindstrm)

Values for water holding in snow and refreezing: Sorry, but these values just make no (physical) sense. Actually, with your value of the water holding capacity basically the snow will hold almost no liquid water, so you probably could skip all the refreezing. Publishing such values could easily cause confusion and should be avoided. Quickly looking Osuch 2015, I could not find the conclusion on these parameter values there. Please clarify!

**Answers to the Editor's comments:**

Dear Editor,

Thank you very much for the comments. We hope that we managed to answer all Your questions, in particular that related to the parameter values. It has a bit philosophical sense and we think there are no simple answers.

1. Eq. 1 and 2 are now changed and all the other multi-letter variables are also corrected.
2. The references were checked and the mistake was corrected.
3. We checked again the values of WHC and CFR parameters.
   The values given in Table 2 were obtained as a result of the optimization of model parameters. These parameters are difficult to identify due to the lack of relevant snow observations. The analysis of sensitivity of these parameters was presented by Benninga (2015) in his master dissertation for the same catchment and it confirms the Osuch (2015) findings. He showed that both parameters have very little influence on the calibration process. The values obtained by him were slightly different than those presented in Table 2, as the calibration period and the objective functions used were different. Still the values obtained by Benninga do not correspond to the default values used in the HBV. The comment was added together with the reference (page 9, lines 22-26:

[revised manuscript text omitted]